# Blood–Brain Barrier Breakdown in Neuroinflammation: Current In Vitro Models

**DOI:** 10.3390/ijms241612699

**Published:** 2023-08-11

**Authors:** Sarah Brandl, Markus Reindl

**Affiliations:** Clinical Department of Neurology, Medical University of Innsbruck, 6020 Innsbruck, Austria; sarah.brandl@i-med.ac.at

**Keywords:** blood–brain barrier, neurovascular unit, neuroinflammation, in vitro models

## Abstract

The blood–brain barrier, which is formed by tightly interconnected microvascular endothelial cells, separates the brain from the peripheral circulation. Together with other central nervous system-resident cell types, including pericytes and astrocytes, the blood–brain barrier forms the neurovascular unit. Upon neuroinflammation, this barrier becomes leaky, allowing molecules and cells to enter the brain and to potentially harm the tissue of the central nervous system. Despite the significance of animal models in research, they may not always adequately reflect human pathophysiology. Therefore, human models are needed. This review will provide an overview of the blood–brain barrier in terms of both health and disease. It will describe all key elements of the in vitro models and will explore how different compositions can be utilized to effectively model a variety of neuroinflammatory conditions. Furthermore, it will explore the existing types of models that are used in basic research to study the respective pathologies thus far.

## 1. Introduction

Typically, 98% of small-molecule drugs, and close to 100% of large-molecule drugs, fail to enter the central nervous system (CNS) through the tight barrier of endothelial cells [1]. Moreover, approximately 30% of all the drugs that are specifically developed for treating CNS diseases encounter failure in penetrating the endothelial cell layer when attempting to access the CNS [2]. However, under certain conditions, this physical barrier becomes permeable. This allows both beneficial and detrimental cells, as well as substances to enter the CNS tissue. Neuronal and glial injury arise due to the inflammatory environment that results from the breakdown of the usually highly effective blood–brain barrier (BBB). Studies investigating the BBB in different neurological disorders in vivo have only been partly successful so far. These animal models did not always reflect the human (patho-) physiology, and the results of animal studies cannot always be replicated in humans. To circumvent this problem, but also to reduce and replace animal experiments, human tissue in vitro models are still urgently needed. This review provides an overview of the current approaches in terms of modeling the neuroinflammation of the BBB in different neurological and psychiatric diseases.

## 2. The Neurovascular Unit in Health

There are different definitions of the vascular BBB: they vary from just the brain microvascular endothelial cells (BMECs) with pericytes in the basement membrane (BM), astrocyte endfeet enwrapping the capillaries, and other cells signaling the formation of barrier features, but not themselves participating in the physical barrier, to the inclusion of the BM and glycocalyx. In contrast, the neurovascular unit (NVU) has become a collective term for all cell types that are involved in BBB integrity, including microglia and neurons.

### 2.1. Microvascular Endothelial Cells

BMECs are the key regulators of the brain microenvironment, and the most important component of the BBB. They cover all brain microcapillaries with a total surface of between 12 and 18 m${}^{2}$ in an adult human [3]. Moreover, BMECs express crucial factors regulating the permeability of the BBB. Microvascular heterogeneity results from the diverse functions that the cell layer needs to exhibit in different tissues. For example, the glomerular endothelium of the kidney is fenestrated with intracellular pores for rapid filtration, absorption, and secretion, whereas the sinusoidal endothelium of the spleen or liver is discontinuous with larger pores for large solute exchanges at higher rates [4,5]. In contrast, BMECs exhibit a continuous, tight layer with the help of gap-, adherens-, and tight junctions (TJs) that bring them into proximity and limit paracellular transport across this layer [6,7]. Adherens junctions consist of transmembrane cadherin–cadherin complexes between adjacent cells, and they are responsible for the adhesion between cells, catenins, and scaffolding proteins in the cytoplasm. Vascular endothelial-cadherins mediate the connection between two BMECs, whereas N-cadherins establish the connections with pericytes [8]. Gap junctions are composed of tissue-specific connexins that form hexamers at the membrane. The alignment of two neighboring endothelial cell–surface hemichannels allows for intercellular communication through the exchange of ions and small molecules between BMECs [7,9]. TJs form the physical separation between the bloodstream and the CNS tissue. They consist of the transmembrane proteins occludin, claudins (especially claudin-5), and junctional adhesion molecules [10,11]. Their extracellular domains are in close interaction with their counterparts on neighboring cells, thereby narrowing the paracellular cleft [12]. The carboxyl-terminal of these transmembrane proteins interact with the actin cytoskeleton and scaffolding proteins, such as zonula occludens-1 and -2 (ZO-1, -2). ZO-1 also establishes interactions between adherens- and gap junction proteins [7].

Many reports from the literature state that only small lipophilic molecules with a molecular weight below 400–600 Da and with a few (<8) hydrogen bonds, as well as gases, can diffuse freely across the BBB. However, there is evidence that there is no distinct cutoff, but rather a molecular weight penalty. There are certain molecules with a higher molecular weight that can diffuse through the BBB by lipid solubility, the largest known being cytokine-induced neutrophil chemoattractant-1 that has about 7 kDa. Furthermore, some substrate classes, such as anti-helminthics, (opiate) peptides, and their analogs, can also cross the BBB by diffusion despite higher molecular weights [2,13,14]. Additionally, diverse transcellular transport systems for other molecules ensure a physiological brain pH and metabolism [15]. Nutrients, such as glucose and amino acids, need solute carrier-mediated transporters to enter the CNS. Many of them have a specific direction and substrate precision. Hormones, organic anions and cations, amines, nucleotides, vitamins, and fatty acids are also carrier-mediated transported [16]. Receptors mediate the transport of protein ligands, such as amyloid-beta (Aβ), transferrin, insulin, and apolipoprotein E [17,18]. Ions are transported across the endothelium via adenosine triphosphatases, uniporters, exchangers, and symporters [15]. The active efflux of drugs (or drug conjugates), xenobiotics, or nucleosides is mediated by adenosine triphosphate-binding cassette transporters. The most prominent ones to name are P-glycoprotein (P-gp) and the multi-drug resistance protein-1 [6]. To conclude, many proteins and other macromolecules cannot cross the BBB, and they are held back in the bloodstream under physiological conditions.

Basolateral BMECs are embedded in the BM. This extracellular matrix (ECM) consists primarily of structural elements, especially collagen type IV; specialized proteins such as laminins, nidogen, fibronectin, and proteoglycans like perlecan; and agrin [19]. Different α- and β-integrin receptors, which form transmembrane heterodimers, provide the functional link between the BMEC cytoskeleton and the ECM. Additionally, the BM has an impact on the integrity of endothelial junctions [19,20].

Toward the vascular lumen, BMECs express the glycocalyx, which is a thin layer of a villiform substance [21]. Its major components are proteoglycan protein polymers and glycosaminoglycan chains, including heparan sulfate, chondroitin sulfate, hyaluronic acid, and their associated binding proteins [22]. The glycocalyx is important for many of the physiological functions of the BBB. Among them, it maintains the low permeability of the BBB, prevents inflammation triggers, and the coagulation response [21]. Furthermore, it was shown to sense changes in the shear force of the blood flow, subsequently inducing the release of endogenous vasoactive mediators [23]. As it is negatively charged, the glycocalyx forms an electrostatic barrier for negatively charged molecules, proteins, and plasma cells [24]. During inflammation, the glycocalyx sheds off the BMECs to enable leukocyte binding to vascular cell adhesion molecules [21]. Furthermore, glycocalyx degradation decreases the physical barrier, enhances the permeability, promotes inflammation by a direct interaction of the BMECs with plasma components or blood cells, as well as interferes with various receptor functions, such as lipid mediators [24,25]. Alterations in the glycocalyx have been shown to affect the BBB integrity, which could promote the development of a broad range of neurological diseases [26].

### 2.2. Pericytes

Pericytes are present in most of the non-epithelial tissues around vessels. However, they are most abundant in the CNS, especially in the retina, where they cover approximately 30% of the vessel surface with varying frequencies depending on their location [27,28]. They are embedded in the BM, which is where they are in close association with BMECs at a distance of less than 20 nm. The membrane facing the microvessels expresses N-cadherin and connexins, which bring the two cell types into a closer proximity. Through their close interaction, the two cell types can exchange ions, second messengers, metabolites, and ribonucleic acids [8]. The distinct functions of pericytes have been discussed rigorously, with contractile abilities that resemble those of smooth muscle cells being the most significant function [29]. Furthermore, they play an important role for the BBB regarding integrity, thereby supporting angiogenesis and microvascular stability [30]. This becomes especially evident in the platelet-derived growth factor receptor-β deficient mice, which do not develop pericytes. These lead to microvascular reductions and microaneurysms. The platelet-derived growth factor is secreted by BMECs, and this leads to the recruitment of pericytes during angiogenesis and, subsequently, vessel stabilization [8,31]. Cerebral autoregulation might also be mediated by pericytes as they have been shown to express receptors for angiotensin I, vasopressin, or endothelin-1 [32,33,34]. A recent review discusses the functions of pericytes in the heart and brain, and shows there are still uncertainties up to today [29]. Injured or degenerating pericytes have been reported in various studies of neurological diseases, such as in Alzheimer’s disease (AD), Amyotrophic Lateral Sclerosis (ALS), stroke, or mild dementia [12].

### 2.3. Astrocytes

Astrocytes received their name from their primarily star-shaped morphology. As the most prevalent cell type within the CNS, these glial cells have more functions beyond just providing support and structure. They are involved in synaptic formation, maturation, plasticity, and neurotransmitter recycling [35,36]. Furthermore, their endfeet wrap around almost the entire outer surface of the brain capillaries, and it is there that they are in close interaction with BMECs. They secrete essential factors for BBB maintenance and are involved in nutrient-waste exchange. Astrocytes enhance P-gp and glucose transporter protein-1 expression in BMECs, as well as in metabolic enzymes [37]. Specialized transporters and pumps regulate the CNS pH, fluid homeostasis, and electrical potential. Prominent examples are the water channel aquaporin-4 (AQP-4) and the inwardly rectifying K+ channel subunit 4.1 [38,39,40]. Astrocytes have been shown to harbor great anti- but also pro-inflammatory potential, which is described in further detail in Section 3.1 [41,42,43,44].

### 2.4. Neurons

The understanding of the importance of neurons for BBB integrity has evolved over the last few decades. They are not involved in BBB formation during early brain development [45]. Despite their approximate distance of 20 µm to the microvessels, they communicate their need, via astrocytes, for additional oxygen or nutrients [46,47]. Astrocytes, in conjunction with pericytes, influence the vascular tone and blood supply in the area to restore physiological conditions. Moreover, glutamatergic neurons can modulate BBB integrity directly by increasing levels of glutamate, as well as influence the BBB efflux transporter gene expressions and endothelial circadian genes [48].

### 2.5. Microglia

Microglia are often referred to as the immune cells of the brain. In a resting state, microglia monitor the brain microenvironment, and they are always prepared to sense antigens via their major histocompatibility complexes [49]. They have important functions in both the adult and developing brain [50]. The phagocytic activity of microglia is crucial for normal brain development by their removal of defective synapses and in helping with synapse pruning [51]. Their role after injury or infection is described below in Section 3.1.

### 2.6. Oligodendrocytes

Oligodendrocytes belong to the macroglia and contribute as cells of the NVU to the BBB. Oligodendrocyte precursor cells (OPCs) have crucial functions in BBB maintenance and vessel formation throughout life. In return, they receive trophic support [52]. By producing and maintaining the myelin sheath as an insulator around axons, they contribute to the critical transmission of nerve impulses. In addition, OPCs are known to communicate with other cells in the BBB, such as astrocytes and pericytes. For example, pericytes stimulate OPCs via laminin-α2, which provokes the differentiation of OPCs into major oligodendrocytes at the sites of damaged, demyelinated axons [53].

## 3. Main Players in Neuroinflammation

### 3.1. Cellular Components

Both brain resident cells, such as glial cells, and peripheral immune cells contribute to neuroinflammation. Upon activation, microglia are polarized into a pro- or anti-inflammatory phenotype depending on the stimulus [51]. The so-called M1 phenotype is involved in the damage of surrounding neuronal and glial cells by secreting neurotoxic factors, such as pro-inflammatory cytokines and chemokines like interleukin (IL)-6, tumor necrosis factor-alpha (TNF-α), C-C motif ligand (CCL)-2, superoxide, and prostaglandin-2 [54]. The M1 type is activated via the classical pathway by pro-inflammatory stimuli, such as interferon-γ, the lipopolysaccharide (LPS) of gram-negative bacteria, or aggregated pathogenic proteins (Aβ, α-synuclein and others) [49,51,55]. The M2 phenotype is involved in tissue repair and wound healing by secreting anti-inflammatory mediators, such as arginase-1 or chitinase-3. This phenotype can be induced by IL-4 or IL-13 in the alternative pathway, or via acquired deactivation by IL-10 or the transforming growth factor-beta (TGF-β). While both phenotypes are in homeostasis during acute stimulation, the pro-inflammatory phenotype is predominant in chronic inflammation. Therefore, excessive microglia activation can lead to a potentiating of tissue damage through a positive feedback loop. This is the case in many neurodegenerative diseases, such as Parkinson’s Disease (PD), AD, or ALS [56].

Similar to microglia, activated astrocytes also exhibit neuroprotective or neurotoxic phenotypes. Liddelow et al. [57] proposed that, depending on the activation trigger, astrocytes develop different entities that are comparable to the M1/M2-type microglia. A1 astrocytes rapidly develop after acute CNS injury, such as CNS brain trauma or neuroinflammation. In response to the pro-inflammatory mediators that are secreted by M1-type microglia, they induce a secondary inflammatory response [49]. This A1-type astrocyte secretes neurotoxic factors that induce the rapid death of neurons and oligodendrocytes, thereby driving neurodegeneration and disease progression [57]. Moreover, it sustains a feedback loop that promotes further M1-type microglia, as well as leads to ECM and TJ degradation via matrix metalloprotease (MMP) and vascular endothelial growth factor (VEGF)-A secretion [58]. In contrast, in ischemia A2, astrocytes have a neuroprotective function, aiding neuronal survival and tissue repair [57]. The A1/A2 nomenclature, however, has been criticized by some researchers due to the high heterogeneity of astrocytes, which extends beyond the binary nomenclature [58,59,60].

Adhesion molecule expression is upregulated in BMECs during inflammatory conditions so as to allow T cells to cross the BBB. P/E-selectins are involved in the initial step, which is also called T cell rolling. The vascular cell adhesion molecule-1 and intercellular adhesion molecule 1 lead to the arrest of CD4+ T helper (Th) cells, or other immune cells, in the processes of capture, rolling, integrin activation, adhesion/arrest, crawling, and diapedesis that occur across the BBB. In autoimmunity, it was shown that additional/added melanoma or activated leukocyte cell adhesion molecules control T cell trafficking into the CNS [12]. More detailed reviews of immune cell transmigration across the BBB can be found elsewhere [61,62,63,64].

### 3.2. Soluble Components

Cytokines mediate the communication within the immune system. Depending on the specific cytokine and the receptor on the receiving immune cell, they can signal a particular pathway for differentiation (such as Th1, Th2, or Th17). Chemokines are released by a variety of cells, including endothelial cells. They play a role in embryonic development and, depending on the chemokine, they recruit specific immune cells. The role of cytokines and chemokines in neuroinflammation and the BBB has recently been reviewed, and the reader is referred to these excellent review articles for more details [65,66,67,68].

MMPs are a group of endopeptidases that are secreted by various cell types, including members of the NVU [69]. Physiologically, they are important contributors in CNS development, angio- and neurogenesis, as well as synaptic plasticity, learning, and memory [69,70]. Among others, MMP-2, -3, and -9 are engaged in BBB degradation and neuroinflammation [70,71]. They are activated through the proteolysis of the N-terminal pro-domain via reactive oxygen species (ROS), IL-17, IL-1β, or TNF-α. In an active state, they degrade ECM and TJ proteins with the subsequent activation of pro-angiogenic factors, such as VEGF [72,73]. Additionally, they further promote a pro-inflammatory microenvironment by cleaving more pro-MMPs and pro-forms of IL-1β or TNF-α [71]. Astrocytes, pericytes, and microglia increase the expression of MMP-9 upon brain injury by increased ROS through the albumin release or LPS [71,74,75]. Additional pro-inflammatory cytokines are produced by the microglia that are activated via MMP-3 by apoptotic neurons [76]. The importance of different MMPs has been observed in a broad spectrum of pathological BBB alterations, ranging from proteolysis after injuries, stroke-associated dementia and hyperglycemia, multiple sclerosis (MS), over epilepsy, and many others [69,70,77,78,79,80].

Lipid mediators are rapidly synthesized in multistep enzymatic pathways, and they are bioactive lipids with functions in both health and disease. Depending on their structure, they are classified into eicosanoids, lysophospholipids, and others. After synthesis, they are released into the extracellular space, where they stimulate BBB cells and modulate their functions via G-protein-coupled receptors. Through the activation of intracellular signaling pathways, the paracellular permeability is ultimately increased [81].

ROS comprise a group of chemically reactive molecules that are involved with oxygen (such as superoxide anions, hydrogen peroxide, and hydroxyl radicals), and they are produced by several endo- and exogenous processes. They have the capability of oxidizing or damaging biological molecules, and they have a dual role in the NVU [82,83,84]. Low levels of ROS, together with reactive nitrogen species, act in signaling transduction for normal cell function maintenance [85]. ROS also contribute to the regulation of blood flow by vasodilation, the clearance of damaged cells, and tissue repair. Maintaining ROS level balance underlies a complex interplay with cellular antioxidant systems, such as the enzymes superoxide dismutase, catalase, and glutathione peroxidase [86]. Imbalances in this regulation cause higher ROS levels, increased blood flow resistance, decreased nitric oxide bioavailability, as well as increased apoptosis and immune response [83]. ROS overproduction has also been associated with several neurodegenerative diseases, especially AD. ROS are involved in several aspects of disease development, including BBB breakdown (which interferes with brain energy supply and homeostasis),and increasing Aβ peptide deposition in vascular walls [86]. ROS promote a senescence-associated secretory phenotype, which involves the secretion of several pro-inflammatory cytokines, MMPs, and insoluble proteins/ECM [84].

## 4. In Vitro Modeling

Not every model is suitable for each research approach. Before starting to develop a BBB in vitro model, the following questions should be considered:What is the research question and the purpose of the experiment?What basic biological requirements does the model need to fulfill?What cells and what ECM are needed for the assays that are to be performed?What time can be invested in each approach?Does it need to be applicable for high-throughput screening?Are there additional devices that are needed for the model?How long are the cells cultured, and when is the endpoint?

An additional overview of the factors that need to be considered for in vitro modeling is illustrated in Figure 1 [87].

### 4.1. Cell Types and Sources

The choice of cell types for each in vitro model is critical and depends on the researcher’s needs and resources. Cells can be derived from animals (especially rodents, but bovine and porcine tissues are also utilized) or humans. In general, they can be distinguished between immortalized cell lines, primary cells, and stem cells. Each cellular model has certain pros and cons. Primary cells are isolated from postmortem brain tissue (or from donor tissue that was collected during a surgical procedure), and they are then frozen in a “low passage”. Thus, postmortem tissues are easier to acquire, but the material isolated from surgeries often serves a better yield [88]. They exhibit characteristics close to the in vivo NVU. However, they depend on a relatively high amount of cell-type-specific growth factors, and they can only divide a limited number of passages before reaching senescence, which affects cellular morphology and functionality [89]. Furthermore, isolation of these cells raises ethical problems and requires high-level technical skills; moreover, purchasing primary cells can be expensive [90]. Therefore, the immortalized cell lines of primary cells were established to tackle this drawback. They possess most cell-type-specific functions, are cost-effective, applicable for long-term culture, and can be expanded rapidly. Unfortunately, immortalization can interfere with morphology and barrier formation. Thus, different cell lines from the same cell type can have aberrant characteristics, which makes it crucial to validate their features (such as cell-type-specific gene/protein expression and barrier integrity before performing experiments [91,92]).

Over the last few years, an increasing number of in vitro BBB/NVU models have been based on stem cells. The opportunity to differentiate them into various cell types with healthy or pathological phenotypes has facilitated new perspectives for investigating the NVU in health and disease [93]. There are several approaches for the use of stem cells as neural stem cells have a low potential for self-renewal and show immune incompatibility upon transplantation [94]. Therefore, they have been used to model diseases by inserting certain gene mutations, and they are then re-injected into animal models rather than being used for BBB models [95]. Mesenchymal stem cells are isolated from bone marrow. Due to the secretion of growth factors that contribute to neuroregeneration and remyelination, they have already been used to treat CNS diseases. Furthermore, since mesenchymal stem cells express similar phenotypic markers as pericytes, they have successfully been implemented in in vitro models as substitutes for primary pericytes [96,97,98]. However, they might be not applicable for neuroinflammatory BBB in vitro models because they have been shown to act immunosuppressively, and they might have AQP-4-regulating functions [96,99]. Finally, induced pluripotent stem cells (iPSCs) are somatic cells that have been reprogrammed to a state of pluripotency by utilizing an overexpression of specific transcription factors [100]. To achieve in-vivo-like paracellular barrier properties, iPSC-based human BBB models are the models of choice; this is because they have a higher transendothelial electrical resistance (TEER) than can be achieved in models utilizing other BMECs [101]. Furthermore, gene-editing techniques, such as CRISPR/Cas9, enable for disease modeling through using iPSCs [102]. Lippmann et al. [103] provided a detailed and widely used protocol for the differentiation of iPSCs into induced brain microvascular endothelial cell-like cells (iBMEC) for the purpose of BBB modeling. This protocol has been modified and further optimized over the last decade [104,105,106]. However, attention needs to be paid to the cell identity, as a recent paper revealed that these differentiation protocols produce a homogeneous epithelial cell population instead of an endothelial one, despite exhibiting a high TEER. Fortunately, this phenotype can be rescued by the overexpression of appropriate BMEC-specific ETS transcription factors (such as *ETV2*, *ERG*, and *FLI1*). Although rescuing the phenotype, Lu et al. [107] stated that, for reliable BBB-forming BMECs, more work needs to be conducted, including thorough characterizations with the latest technologies. Moreover, the narrow experimental window caused by de-differentiating iPSCs under in vitro conditions results in high costs for culturing, and time-intensive procedures pose limitations in iPSC models [90]. However, as an aspect of personalized medicine, patient-derived iPSCs are becoming increasingly important. In addition to being used for modeling diseases, these cells can also be helpful for the individual assessments of putative drug responses [93,100,102]. For example, in monogenetic diseases, such as Huntington’s disease (HD), the mutations are already present in patient cells and do not need to be created additionally [108,109].

Undoubtedly, BMECs are the most fundamental compartment of the BBB. Therefore, culturing them as a monoculture can be utilized as a BBB model. However, as mentioned above, other cell types crucially contribute to BBB integrity [104]. To date, due to their simplicity, there are still monoculture models in use. These cultures are utilized in basic research for toxicity and proliferation evaluations, transport experiments, and the characterization of the secretion or immune response, especially when the response of only BMECs is aimed at to be studied without the interference of other NVU cell types [87]. Additionally, it needs to be noted that BMECs have a regional heterogeneity and an arterial-capillary-venous zonation in vivo, which could lead to varying results in in vitro experiments depending on the utilized BMEC source [110]. Very apparent differences in morphology and function are visible in astrocytes and BMECs between gray and white matter, such as the expression of specific receptors and transporters. The spatial density and orientation of the BMECs vary depending on the location, and astrocytes show fibrous morphology with long processes in the white matter while being more star-shaped in gray matter. Furthermore, oligodendrocytes are more abundant in white matter compared to the gray matter. Pericyte coverage is also heterogeneous along the vessels. Differences have been reviewed in more detail by Villabona-Rueda et al. [111]. Depending on the pathology that is to be modeled, these differences need to be considered. For example, MS patients show different types of lesions and BBB dysfunctions in their gray and white matter, and neurodegeneration in PD and AD primarily affects gray matter [111,112].

### 4.2. Three-Dimensional ECM

In a 2D perspective, the ECM is set up solely by a layer of glial and/or neuronal cells that are seeded onto a BM-mimicking substance. The currently used BM substitutes are primarily collagen type I/IV and/or fibronectin [87]. To take ECM modeling to the third dimension, a functional scaffold that supports appropriate cell growth and differentiation is necessary. It must fulfill certain requirements, including ECM-comparable bio-physiological properties (low stiffness, good hydrophilicity, elasticity, and degradation) and a low toxicity for the cells [113]. Hydrogels are the medium of choice for these demands. They consist of a biocompatible network of cross-linked polymer chains of natural or synthetic origin. Natural polymers are either polysaccharide-based, such as hyaluronic acid or chitosan, or protein-based, such as collagen, laminin, fibrin, or fibronectin [114]. Common substances incorporate several proteins, such as gelatin, Geltrex™, or Matrigel^®^. The latter ones have a thick structure and weak cross-linking, which makes them beneficial for 3D BBB self-assembled models [115]. There are also plant-based materials available, such as, for example, alginate, which is a polysaccharide that is derived from brown algae [116]. Although natural polymers reflect the physiological composition of the ECM, they have several disadvantages, such as batch-to-batch variability, an (often) animal origin, and weak mechanical properties, thereby leading to more rapid disintegration [117]. In contrast, synthetic polymers are chemically defined and highly versatile, but they also poorly reflect the physiological ECM composition. The primary synthetic substance that has been used for BBB modeling is polyethylene glycol. Natural polymers combined with synthetic polymers ensure a balance between mechanical strength and biocompatibility. These hybrid hydrogels can have similarities with different CNS tissue characteristics, since ratios between the constituent polymers, as well as functional groups or cross-linking agents, can be modified. Consequently, their properties, such as degradation rate or stiffness, are tunable, and they can be used to model healthy or pathological neural tissues. Examples of often-used hydrogels are polyethylene glycol-hyaluronic acid, polyethylene glycol-collagen, or gelatin-methacryloyl [118]. All hydrogels need some kind of cross-linking. This can be achieved physically in the case of natural polymers, and this happens without the addition of exogenous agents, which occur solely by the change in temperature or pH. Hydrogels with synthetic compartments need chemical cross-linking, which is typically photo-polymerization by an external substance that induces gelation [119].

### 4.3. Transwell

With the first model being established in 1953, the transwell model setup has not changed substantially since [120]. This approach is still widely used due to its cost-effectiveness and high-throughput ability. In general, transwells consist of a polystyrene multi-well carrier plate, which is available in several sizes, and its respective inserts. The bottom of the inserts consists of a porous membrane that allows for the exchange of nutrients and other molecules between the apical and basolateral compartments of the insert. The pore size and density can vary depending on the application. Usually, between 0.2 and 3 µm pores are used for transport studies and tissue engineering [121,122,123]. Thus, interactions between different cell types can be studied in co-culture models while maintaining a physical separation. Smaller pore sizes are also recommended for the proper formation of BMEC monolayers for barrier function assays [121]. When the chemotaxis or invasion of cells is studied, membranes with larger pores of up to 12 µm are used [124]. The material of the membrane should be chosen individually depending on the application. Polycarbonate membranes are made of a translucent thermoplastic polymer, and they are employed in many cell culture studies due to their ease of use and low cost. However, they are not fully transparent, and visualizing apically seeded cells under the microscope requires cell labeling with fluorescent or chromogenic dyes. Furthermore, they have high protein binding capacities, which is unsuitable for certain types of assays. Polyethylene terephthalate membranes consist of polyester and exhibit low protein binding. They are clear and offer good cell visibility. However, they have limited chemical resistance, which is unfavorable for some applications. Polyvinylidene fluoride membranes have uniform or asymmetric pores, low protein binding, as well as high chemical and thermal stability. They are suitable for a wide range of applications, but are also more expensive. Coating the membranes with BM components, such as collagen type IV or fibronectin, is recommended for better cell attachments and morphologies [123]. Recent models used the co-cultures of human immortalized cell lines [123,125], primary cells [122], or iPSCs [126,127,128,129]. Some of them incorporated hydrogels on one side of the insert with glial cells to obtain a more physiologically relevant ECM setup [129,130,131]. Transwell co-cultures are often criticized for their lack of physiological relevance due to the absence of contact between different cell types. The thickness of the membrane, which is typically at least 10 µm thick, restricts the cell–cell interactions across the membrane. This problem was addressed by Zakharova et al. [132], who developed polydimethylsiloxane (PDMS) membranes with tunable thicknesses and pore sizes. They fabricated an optically transparent, 2 µm thin membrane that enabled for a lower permeability in BMECs when co-cultured with astrocytes when compared to other membrane substances.

### 4.4. 3D Models

Three-dimensional models have made some advances compared to the classical transwell models. First, a 3D ECM provides a more in-vivo-like environment than the planar cell layers of most transwell models. Second, aside from membrane-based microfluidic models, the different cell types are in direct contact with each other, which facilitates the interaction and exchange of secreted growth factors or other regulatory molecules. Third, a medium flow mimicking the shear stress of the blood circulation can be implemented.

The neuroinflammatory response profile to TNF-α stimulations in 2D vs. 3D models was investigated by Herland et al. [133] in mono- and co-culture models by utilizing transwell and microfluidic BBB-on-a-chip setups. Co-culturing primary BMECs with astrocytes and pericytes under flow conditions resulted in significantly higher cytokine secretion (i.e., of IL-6, the granulocyte colony-stimulating factor) in the 3D model. Cucullo et al. [134] showed, in 2011, the importance of shear stress on BMECs, as it altered the gene expression patterns of junctional proteins, CYP450 proteins, ion channels, drug transporters, adhesion molecules, and integrins. Notably, evidence has arisen of the notion that shear stress does not affect cell morphology, but rather it tightens the barrier itself by an increase in adherens junction and TJ protein expression [87]. The diameter of the vessel (channel, tubing), the dynamic viscosity of the fluid (medium), as well as the flow rate need to be adjusted to get close to the physiological amount of shear stress that is needed in the model. The in vivo shear stress is approximately 0.3–2 Pa (4 to 30 dyn/cm^2^), and this is with a dynamic viscosity, which is dependent on the rheology of blood and is usually calculated as 3.5–5.5~cP [87,135,136]. The calculation of the shear stress adjustment is described in detail elsewhere [87,137,138]. For medium-flow generation in microfluidic devices, syringe and peristaltic pumps are widely used in several other methods [139]. However, there are new advances in pumping systems that passively pump media in a non-mechanical manner [140]. It is important to note that this is a continuous rather than pulsatile flow, as is keeping the shear stress on a moderate, physiological level [141]. Faley et al. [142] created a 3D microfluidic model with a tubular structure filled with the monolayers of endothelial cells. The authors compared different endothelial cell lines, ranging from human umbilical vein endothelial cells to human dermal microvascular endothelial cells, as well as to two iPSC-derived BMEC lines. The iBMECs showed a 10–100 times lower permeability to different-sized markers compared to the other cell lines. Moreover, the model maintained the barrier function and efflux transporter activity for up to 21 days under perfusion conditions. It is worth mentioning that Faley et al. saw the best results with a subphysiological wall shear of 0.3 dyn/cm^2^. Elevated flow rates led to an increase in permeability and angiogenic sprouting.

One common problem regarding microfluidics is the formation of bubbles, which are generated when plugging or removing pipes, exchanging reagents, or if the systems are not fully tight. These bubbles tend to accumulate at right angles in the microfluidic chips, particularly at the device inlets. Moreover, these bubbles cause blockages in the channels known as “dead zones.” As such, they lead to flow interference, which increases the pressure and causes cell membrane damage. To address this problem, active (for example lasers or acoustic generators) and passive (which do not require additional equipment) de-bubblers can be used [143].

#### 4.4.1. Microfluidic Models

The first microfluidic models were described in the late 1990s, and they consisted of a hollow fiber cartridge system that was composed of a bundle of porous polypropylene fibers. Through the fibers, there was a medium flow applied by a variable-speed pulsatile pump. Endothelial and glial cells could be seeded through separate loading ports. The tubing was made of gas-permeable silicon [144,145]. Continuing up to the present day, this original setup has been further developed and optimized, and the research on novel microfluidic devices is still intensive. However, the major components are still present: (1) a carrier in which the cells are incorporated, (2) an ECM compartment, (3) a BMEC monolayer, and (4) medium flow. The material of microfluidic chips has mainly been PDMS since it fulfills most needs. It is biocompatible, easy to fabricate, cost-efficient, and transparent. However, one limitation of the material is its tendency in the absorption of hydrophobic molecules. This problem is minimized by conducting protein modifications at its surface [146].

However, there are versatile possibilities for the chip appearance, and there are four major designs [137]. The sandwich design comprises a PDMS chip with two superposed channels, which are separated by a polycarbonate or PDMS membrane. Glial cells or neurons are seeded into the upper channel, which mimics the CNS matrix. The lower channel reflects the capillary with the BMECs being seeded onto the membrane. Similar to this setting is the parallel design. Cells are seeded not in vertically but in horizontally separated channels. These channels have small interconnecting pipes with a diameter of 3 µm to enable communication between the cell types [147]. As a further development of the parallel design, chips with two channels for BMECs and an additional channel in between with a CNS matrix replicate have been established. This design is membrane-based as well, but incorporates a hydrogel-aided compartment for the ECM [148]. The communication between cells can be further enhanced by 3D tubular structure designs. Without the need for a membrane, the cells are in direct proximity. Cylindrical channels lined with BMECs are surrounded by a cell-containing hydrogel, thus reflecting the ECM. The microvessel structure can either be achieved artificially or with self-assembled microvessels in hydrogels [149]. Campisi et al. [150] combined an iBMEC network with human brain pericytes and astrocytes within a single fibrin hydrogel. After vasculogenesis, the model exhibited physiologically relevant structures and functionality. Organs-on-a-chip microfluidic models in various setups can be purchased as well. Different companies have launched chips for all kinds of research areas. The reviews of Nikolakopoulou and Jagtiani [151,152] sum up the currently available products.

#### 4.4.2. 3D Bioprinting

Over the last few decades, 3D bioprinting has pushed the precision of BBB model fabrication to a new level. Templates are designed by computer-aided design software and then printed by specialized 3D printers. Depending on the printing technology, precise structures with a resolution down to 100 µm (extrusion-based bioprinting) and even 10 µm (laser-assisted bioprinting) are possible [113]. There are different fabrication techniques, each of which imply strengths but also limitations [119]. Inkjet bioprinting offers a high speed and resolution with low cost. It is favorable for printing cells as it ensures a high cell viability of up to 90% [153]. A printer head connected to a cartilage creates droplets of bioink by a thermal or piezoelectric actuator [154]. To prevent the printer head from clogging, printing high cell densities or viscous material need to be avoided [119]. Microextrusion bioprinting allows for the printing of high-viscous material in a continuous strand through a syringe via a screw plunger or through air pressure. Compared to other bioprinting methods, it is relatively cheap and simple [154]. However, it compromises cell viability by exposing them to high mechanical stress. Therefore, extrusion-based bioprinting is used more for scaffold and sacrificial structure printing [155]. In laser-assisted bioprinting, the bioink is suspended at the bottom of a donor layer, which has a ribbon structure on top. The printing of droplets is achieved by an absorbing layer that is stimulated by a laser pulse. This pulse leads to an evaporation of the donor layer, creating a bubble at the interface of the bioink layer, which is pushed onto the substrate. There, the bioink droplet is then cross-linked. This allows for the printing of very precise structures with a broad range of bioink viscosity. Due to the contact-free printing, cells do not experience mechanical stress. Although cell viability is high, the technique’s high costs restrict its use in research [119,154]. Different kinds of light-based bioprinting (stereo-, soft-, and two-photon lithography) are rapid and highly precise methods, and they are conducted by a hydrogel layer being exposed to a light pattern (usually UV or near-UV wavelengths); through this, the illuminated area is cross-linked. This process is repeated vertically until the whole construct is built. Scaffolds can be printed in such precision that Mariano et al. [156], for the first time, developed a 1:1 scale biomimetic BBB model. Two years later, the same group published a model with even thinner capillary walls of only 2 µm instead of 10 µm, as was the case in the prior model [157]. Despite the high cell viability and fast fabrication, a drawback of these methods is the limitation to single-material structures [119,154].

The printable material in 3D bioprinting is highly diverse. It needs to be distinguished between substances that incorporate cells and those that are utilized for frameworks or for dissolving support. Bioinks with cells are hydrogels, as described in Section 4.2. The frameworks for the chips themselves or scaffolds, where cell-laden hydrogels are added in between, are usually made by highly viscous substances that retain their structure. Resins or PDMS are the materials of choice [157,158]. Sacrificial inks can be used to print hollow structures into matrices since they dissolve under specific circumstances, like when there are temperature changes. Examples are Pluronic (F-127) or polyvinyl alcohol [158]. Even the chips themselves can be printed, which is achieved by utilizing transparent polymers like transparent poly methyl methacrylate) or duralumin [158]. Wang et al. [159] developed an Objet VeroClear photopolymer-based chip with a perylene-C coating to ensure transparent appearances, chemical resistance, and biocompatibility.

#### 4.4.3. Cell Aggregates (Spheroids and Organoids)

Spheroids and organoids are both self-assembled cell aggregates in a static three-dimensional environment. However, they differ in composition, complexity, and size. Spheroids are more simple cell structures that are primarily generated by cell lines, primary cells, or tumor cells/tissues in suspension or non-adherent conditions. They do not require an ECM, and they usually do not exhibit a defined tissue architecture. Organoids are derived from culturing iPSCs, embyronic, or neural stem cells, and they can recapitulate many aspects of tissue and organ function through which to assess neurotoxicity and to develop new drugs. However, cell differentiation requires an ECM and a growth factor cocktail [160]. The first approaches were established a decade ago and are progressing rapidly [161,162]. For example, Nzou et al. [163] studied the suitability of their human cortex organoid model for drug development by assessing the BBB impairment under hypoxic and neuroinflammatory conditions, as well as the impact of ROS and inflammation-reducing reagents. One major advantage of organoids is the possibility of modeling different brain regions. Thus, organoids resembling the human midbrain can be developed, including astrocytes, oligodendrocytes, and dopaminergic neurons that are myelinated and exhibit synaptic connections [162]. Furthermore, the self-assembled structures can be set up based on patient iPSCs, which allows for disease- and patient-specific modeling. In using this approach, the potential neurotoxicity for the patient can be screened and might also be helpful in therapy development in the future [163]. A disadvantage is that organoids are limited in size (around one millimeter) due to the lack of nutrient availability in their core. This often leads to cellular stress within the CNS-mimicking organoid center, resulting in necrotic cells with fragmented nuclei. This issue could be solved by vascularization within the organoid. However, this is still a major challenge for researchers in this field [164]. A study by Wörsdörfer et al. [165] suggested the implementation of mesodermal progenitor cells into organoid models in order to tackle the lack of vessels within the 3D structure. Their tumor organoid model incorporated blood vessels, as well as BM and cell–cell junctions. Additionally, the mesodermal progenitor cells also differentiated into microglia-like cells that could potentially be used for tumor modeling but also neuroinflammation in general. Ham et al. [166] developed vascularized cerebral organoids by adding VEGF from the beginning of the differentiation of embryonic stem cells, thus creating open-circle vascular tubes with a two-layer structure and BBB characteristics without disturbing neurogenesis. However, in long-term cultures, the density of blood vessels decreased and four-month old organoids had distorted open-circle morphologies, which was possibly caused by the missing blood pressure. Therefore, the importance of shear stress within, as well as on, the outer surface of cerebral organoids is still a matter of research [89,166].

### 4.5. Model Validation

#### 4.5.1. Cell Identity and Viability

The validation of the cell-type identity is not only for the iPSCs of special importance. Although manufacturers promise the expression of their commercially immortalized and primary cell lines, they still might change their functions and morphology due to immortalization or cell culture conditions, as well as due to the number of passages. Therefore, cell-type-specific markers should be assessed via immunofluorescence, Western blot, qPCR, or flow cytometry before the performing experiments [119]. BMEC identity is usually assessed by a combination of general endothelial cell markers—such as vascular endothelial-cadherin (CD31), specific markers for functionality, and TJ protein expression—which is a topic that will be covered in the next section. Astrocyte markers are primarily GFAP and AQP-4. While GFAP can also be found on other glial cells in the development of natural tissues, it is expressed only by astrocytes in a differentiated neuronal culture [167]. AQP-4 is most abundantly expressed by astrocytic endfeet, but other ependymal cells show low expressions as well [168]. Both markers can be instrumental in disease modeling as their expression enhances or decreases in the course of pathogenic changes or through the reactivity of astrocytes [167]. Pericyte identification can be challenging if they are isolated from tissues with vascular smooth muscle cells, but this can be neglected in the microvasculature, as this cell type is not usually found there. The platelet-derived growth factor receptor-β and neural/glial antigen-2 are accepted markers for pericyte identity [169]. The oligodendrocyte mature markers are oligodendrocyte transcription factors 1 and 2, or myelin-associated proteins, including myelin basic protein, 2′,3′-cyclic nucleotide-3′-phosphodiesterase, as well as myelin oligodendrocyte glycoprotein (MOG) [119,170]. The latter is of special importance for the investigation of inflammatory demyelinating diseases. Depending on their reactive state, specific microglial markers can be used. Resting microglia express CD11b and CD45. When entering a pro-inflammatory phenotype through external stimuli, M1 microglia express CD86 and inducible nitric oxide synthase, whereas anti-inflammatory M2 microglia upregulate their arginase-1 and CD206 expression [171]. Furthermore, the development, maturation, and functionality of neural cells can be investigated in 3D fabrications. For neural differentiation, nestin is often used as a progenitor and as a neural stem cell marker [172]. The neuron-specific class III β-tubulin and its antibody TUJ are markers for premature or undifferentiated neural cells. Major neurons can be identified by microtubule-associated protein 2. Mature neuronal subtypes can be further divided by the markers gamma-aminobutyric acid (GABA) and its enzymes GAD65/67 for interneurons; tyrosine hydroxylase for dopaminergic neurons; tryptophan hydroxylase for serotonergic neurons; and choline acetyltransferase for cholinergic neurons [119].

Assessing the cellular viability is particularly important in 3D models since nutrient and oxygen availability is not always ensured; moreover, it is dependent on the thickness and porosity of the matrix and the chip itself. Furthermore, extensive UV cross-linking during fabrication is compromised with cell viability. Besides checking cell morphology and the counting tools that are available for confocal microscopes from 3D stacked image sequences (such as ImageJ plugin 3D Objects Counter), several reagents can be used for viability evaluation [173]. Tetrazolium salts (like MTT) are reduced by living cells into formazan dyes, which can then be quantified via visible light absorbance measurements. Likewise, resazurin-based substances, such as Alamar Blue, can be used for estimating the living cell population, whereby the number of living cells is estimated through reduction reactions that produce the red fluorescent resorufin. The advantage of tetrazolium- and resazurin-based chemicals is their simple application to the culture medium, which is incubated for a defined time and can be measured immediately afterward. However, these methods do not reflect the dead-cell compartments, and reductions in a 3D ECM occur only slowly [174]. For the evaluation of both viable and dead cells, a mixture of Calcein-AM (which can easily enter living cells) and ethidium homodimers (which only enter dead or damaged cells) is the combination of choice, if it is applied with different fluorescent tags [119]. Apoptosis can be evaluated in assays with markers like caspase-3 and -9, as well as Bcl-2 and Bax, or with terminal deoxynucleotidyl transferase dUTP nick end labeling assays. Since apoptosis is detectable in diverse pathological conditions, such as traumatic brain injury (TBI) or ALS, assays should be also included in the in vitro modeling of these diseases [175,176].

#### 4.5.2. TJ Formation and Permeability

BBB permeability is most frequently tested by TEER measurements. The presupposition is that the barrier has a setup that allows for measurements on the basolateral and apical side. As such, an electrical field is created at both sides of the barrier, and the resistance is measured by electrodes. The higher the resistance, the stronger the barrier function and the more tightly the cell junctions are connected. The advantage of this method is the non-invasive and rapid manner in which the barrier can be assessed.
TEER=R∗A
where *R* depicts the electrical resistance in Ω (which is equal to the measured resistance minus the resistance of the blank) and *A* describes the growth area in cm^2^. The average physiological electrical resistance in vivo is around 1800 Ω cm^2^, but this can reach values as high as 8000 Ω cm^2^ [177,178]. However, only very few current in vitro platforms can reach values as high as this [159,179,180]. Although TEER measurement is the gold standard for permeability assessments in transwell and microfluidic model systems, the technique lacks consistency in the results. A critical review from Vigh et al. [181] highlights the range of values that depend on the device used for measurements. The authors stated that the correct interpretation of values, as well as the comparison between the models, is only possible with an explicit description of the technical parameters and the setup. Palma-Florez et al. [182] tried to address this limitation by developing a BBB microfluidic chip with an integrated micro-TEER device that was in a close proximity to the barrier. To improve TEER, certain substances can be added. For instance, synthetic glucocorticoids like dexamethasone or hydrocortisone (cortisol) have been found to increase TEER values in vitro. However, they may impact experimental results, particularly in studies that are related to inflammation as glucocorticoids have anti-inflammatory properties [183,184].

Permeability can also be assessed by specific markers that are non-toxic and do not bind, or through those that become internalized by the cells within the model. Furthermore, they should be metabolically inert, available in different molecular sizes, quantifiable, and reliable [12]. Detailed descriptions of how permeability needs to be calculated in 2D and 3D setups were published by, for example, Hajal et al. [185] and Wong et al. [186]. Kadry et al. [12] provided an overview of the different reagents that can be used to test permeability within the context of various approaches.Since none of the substances that are currently available fulfill all the aforementioned requirements, the choice of marker needs to be made based on the specific research question. Both small (for examining small changes) and large molecules (to assess BBB integrity loss) can be used as markers for BBB dysfunction. Since none of the substances currently available fulfill all the necessary requirements, the choice of marker for BBB dysfunction must be made based on the specific research question. Small molecules are suitable for studying small changes, while large molecules can be utilized to evaluate BBB integrity loss. As such, a combination of different markers is the most reliable approach through which to evaluate BBB integrity [12,187]. TJ proteins can either be directly evaluated via immunofluorescence, qPCR, or via Western blotting. The most commonly accepted markers for the BBB are claudin-5, ZO-1, and occludin. Additionally, the markers for efflux transporters, such as P-gp, BCRP, Mrp, or solute carriers can be utilized to ensure BMEC functionality and polarity [187]. A recent study by Nakayama-Kitamura et al. [123] set up evaluation parameters for the in vitro human BBB likeness that is needed for drug development. The parameters comprised TJ markers (claudin-5 and ZO-1), TEER, the endothelial cell marker CD31, transporters (P-gp, glucose transporter protein-1, and BCRP), and receptor-mediated transcytosis (transferrin receptor). Additionally, they checked the permeability of caffeine (which is usually BBB-permeable) and Lucifer Yellow (which is impermeable if the BBB is functional).

## 5. Disease-Specific Modeling of Neuroinflammation

The following section will discuss the various conditions that involve neuroinflammation. However, since there are more than 600 neurological diseases known today, it will only cover a fraction of the most prominent examples that come with BBB impairment [188]. Key features of the discussed neurological diseases are summarized in Table 1.

### 5.1. Neurological Autoimmune Diseases

BBB dysfunction is an early event in MS, which is followed by immune cell infiltration and the extravasation of plasma proteins. How the BBB contributes to MS pathogenesis and progression is discussed elsewhere [46,220,221,222,223]. Nishihara et al. [224] showed that different Th subsets had comparable migration rates through the BBB in a transwell setup under inflammatory conditions (mediated by TNF-α and interferon-γ). However, it was observed that the Th1 cells preferentially migrated through the barrier in non-inflammatory conditions. The authors also investigated the movement through the blood-cerebrospinal fluid barrier and saw a 10- to 20-fold higher migration compared to the BBB model, especially regarding Th17 cells. Based on these findings different Th cell subsets might use different anatomical routes to enter the CNS. The Wnt/β-catenin pathway is involved in numerous crucial regulations of embryonic development and adult tissue homeostasis. Among other functions, it is important for maintaining the BBB’s integrity and restoring it in disease, including in the case of neurodegenerative diseases like AD, PD, and ALS [225,226]. In MS, in vivo and in vitro studies have revealed that abnormalities in Wnt/β-catenin signaling contribute to re-myelination failure. OPCs cannot detach from the vasculature and are unable to migrate to the demyelinated region. Furthermore, un-detached OPCs lead to perivascular clustering and the secretion of Wif1, which reduces Wnt and TJ integrity [227]. Derada Troletti et al. [228] showed that endothelial to mesenchymal transition, which they provoked by TGF-β and IL-1β additions, mediated the inflammation-induced BMEC dysfunction, and they found that this might also play a role in MS pathophysiology. The main transcription factor of endothelial de-differentiation, ETS1, has also been shown to be associated with BBB breakdown [229]. Cerutti et al. [230] developed an in vitro technique for studying the interaction between BMECs with human leukocytes in a microfluidic model, which was coupled with live-cell imaging that could also be advantageous for leukocyte extravasation into the CNS in MS in vitro studies. It appears to be impossible to create a universal in vitro model for neuroinflammation in MS due to the reliance of adaptive and innate immune cells on the disease’s development, as well as due to the specific region of the brain that is affected [192].

For drug development, BBB in vitro models are urgently needed to investigate the effects of treatments for retained integrity or barrier restoration. In the CNS, the different pathogenic autoantibodies have different modes of action. Without proper treatment, CNS autoimmunity can have severe effects in patients, leading to cognitive impairment, seizures, or even comas in NMDA-receptor encephalitis (NMDARE), or MS-like relapses in AQP-4 antibody-positive neuromyelitis optica spectrum disorders (NMOSD). Acute relapses are typically treated with general immunosuppression via the use of corticosteroids. Although corticosteroids decrease the permeability of the BBB and the production of pro-inflammatory cytokines, chemokines, and cell adhesion molecules, they can also have severe side effects. Therefore, more targeted therapeutics are needed [231]. Takeshita et al. [232] developed a tri-culture transwell model, whereby they applied the antibodies from NMOSD patients who were with or without the IL-6 receptor blocking therapeutic antibody satralizumab. The addition of satralizumab led to higher TEER values and decreased the transmigration of peripheral blood mononuclear cells through the barrier. When studying the recombinant antibodies from NMOSD patients’ cerebrospinal fluid, Shimizu et al. [233] observed a strong binding in one of their antibodies against glucose-regulated protein 78, which is an endoplasmic reticulum chaperone that has also been found on the surface of BMECs in vitro. Therefore, the hypothesis that arose was that other non-disease-defining autoantibody targets are relevant for the BBB breakdown in these diseases, and that they could either be possible therapeutic targets or even be employed for the enhanced accessibility of immunotherapeutics into the brain. To investigate this hypothesis, Li et al. [194] analyzed the human monoclonal antibodies from patients with NMDARE and GABA_B_-receptor encephalitis for non-disease-defining antibody target identification. Antibodies that showed vascular binding in mouse brain sections were applied to a BMEC monoculture transwell system to evaluate if there were effects on their microvasculature.One of their monoclonal antibodies decreased TEER and occludin expression, which was also confirmed in vivo. Finally, myosin-X was identified as a novel target epitope.

### 5.2. CNS Infections

Many bacterial species are able to infect the CNS with intact, but also especially impaired, BBBs via diverse mechanisms [234]. Brown et al. [235] investigated, in their 3D microfluidic NVU dual chamber model, the impact of inflammatory stimulation on the BBB, and this was induced by LPS (mimicking a systemic bacterial infection) or via pro-inflammatory cytokines (which could result in a local or systemic inflammation). By applying either 100 μg/mL of LPS or a 100 ng/mL cytokine mix (of IL-1, TNF-α, and monocyte chemoattractant protein-1,2) to the vascular compartment, inflammation was induced, and the metabolic profile on both sides was investigated over 24 h of exposure. An activation of pro-inflammatory cytokines was found in both vascular and brain compartments, whereas, at later time points, this activation was only seen in a subset in both compartments. This could suggest the activation of anti-inflammatory cytokines at later time points within the brain section, whereas the vascular compartment remained as more pro-inflammatory. The bacterial infection of the CNS that causes bacterial meningitis has been modeled with different pathogens, and this was achieved with several BMEC cell lines in the transwell models [236]. *Streptococcus pneumoniae* (among others) was shown to upregulate VEGF via hypoxia-inducible factor-1α induction.

Consequently, BBB permeability was elevated, thus enabling more bacterial transmigrations into the CNS [237]. VEGF secretion was also observed by Caporarello et al. [238] when infecting their BMEC-pericyte co-culture BBB model with *Haemophilus influenzae* type a. The authors demonstrated that different adenosine receptors on both cell types were activated upon infection and that VEGF, was released by the cells, which can cause pericyte detachment, BMEC proliferation, and BBB breakdown. More recently, the impact of the gut microbiome on the CNS and BBB has been extensively investigated and reviewed [239,240,241,242]. In line with the “endotoxin hypothesis,” developing multi-organs-on-a-chip platforms could assist in filling in the knowledge gap concerning how the microbiome may contribute to neurodegeneration [243,244,245].

Besides bacteria, fungi can penetrate the BBB, with *Cryptococcus neoformans* being the most common one to cause meningitis. A promising NVU-on-a-chip was established by Kim et al. [246], which was used to analyze the penetration of the BBB by *Cryptococcus neoformans*. The pump-free model comprised a unilateral medium flow, human neural stem cells, BMECs, and pericytes. The authors observed an elevation of inflammatory and angiogenesis-related cytokines, such as IL-8 and thrombospondin-1, but there was no BBB impairment. Therefore, a transcytosis-mediated entry of the pathogen into the CNS was proposed, and the authors concluded that including additional tissues in the model to create a multi-organ-on-chip model would be beneficial. This could help in examining the gut–brain axis and in studying fungal penetration from the gut through the BBB, as well as in verifying its neurotropism.

Over the last few years, the COVID-19 epidemic has led to comprehensive research on severe acute respiratory syndrome coronavirus type 2 (SARS-CoV-2). Since neurological symptoms were reported early on during the acute phase, but also in long-term COVID-19, the transmigration through the BBB and the effects of a severe viral infection of the CNS were extensively studied. Bipolar neuron-to-neuron spread in the olfactory epithelium, transport via the vagal nerve to the brain stem, or transmigration through the blood-cerebrospinal fluid barrier or BBB (directly or on leukocytes) were all under discussion as entry routes for SARS-CoV-2 [247]. Kase et al. [248] observed that the pseudo-typed lentivirus particles of major SARS-CoV-2 strains were able to infect microglia, whereas they rarely infected other CNS cell types such as iPSC-derived neurons and astrocytes. Another study by Andrews et al. [249] produced contradicting results in their stem-cell-derived organoids, with astrocytes being the major target of SARS-CoV-2. Rhea et al. [250] tested the radiolabeled monomeric SARS-CoV-2 spike protein subunit S1 on its ability to cross the BBB in vivo and in vitro. The authors suggested an adsorptive transcytotic manner with the participation of angiotensin-converting enzyme 2, which is the main endogenous receptor of the virus, in a murine brain and lung. Moreover, they noticed a more effective uptake of S1 in all brain regions when it was taken up across the BBB compared to the nasal route. This uptake of S1 was only slightly affected by LPS-induced inflammation, whereas LPS inflammation altered the S1 clearance from blood and its uptake by peripheral tissues. However, these transwell experiments with human iBMECs showed a limited permeability for S1, probably due to technical reasons. Buzhdygan et al. [251] used a hydrogel-based BBB model, which they exposed to the SARS-CoV-2 spike protein subunits S1 and S2. Although the spike protein induced a BBB integrity loss, it did not affect BMEC viability; rather, it enhanced the vascular cell/intercellular adhesion molecule-1, the pro-inflammatory response (i.e., IL-1β, IL-6, C-X-C motif ligand-10, and CCL5), and MMP (especially MMP-3/12) secretion. A study by DeOre et al. [252], who employed a SARS-CoV-2 spike protein in a hydrogel-based microfluidic BBB model, revealed that the angiotensin-converting enzyme 2 expression was altered by the addition of the S1 spike protein subunit when it was paired with fluid shear stress. Furthermore, RhoA was identified as a major regulator of the BBB breakdown through the spike protein. Zhang et al. [253] found an impairment of the BM but not TJs in their co-culture BBB transwell model with BMECs and astrocytes after SARS-CoV-2 infection. The virus was able to replicate in the BMECs; furthermore, it passed the barrier, probably via transcytosis, and degraded the BM (by upregulated MMP-9). Manosso et al. [254] investigated the microbiota–gut–brain communication during SARS-CoV-2 infection and suggested that the SARS-CoV-2-induced cytokine storm leads to microglial activation, astrocyte reactivity, and neuronal degeneration, which then promote the development of psychiatric and neurological symptoms. In a study by Ju et al. [255], it was suggested that the envelope protein is responsible for the breakdown of the BBB during SARS-CoV-2 infection. The transwell model used showed a decrease in cell viability and an increase in inflammatory mediators (major histocompatibility complex-I, IL-1β, and—particularly—IL-6) when the envelope protein was added. Furthermore, the ZO-1 mRNA levels were decreased after SARS-CoV-2 envelope protein administration.

Human immunodeficiency viruses (HIVs), different flaviviruses, and new-world alphaviruses have also been studied for BBB penetration and CNS infection in vitro [201,256]. Several studies showed the ability of the Zika virus and other flaviviruses in terms of infecting and activating BMECs [257,258,259,260,261,262]. Moreover, the virus crossed the BBB via transcytosis without significantly increasing permeability [258,259,260]. However, controversial results from other studies have suggested that TJ breaks down in a virus strain-dependent manner, thereby enabling for a CNS entry via a paracellular route [263]. For HIV-1, it was demonstrated that TJ proteins are disrupted upon infection [201]. Exposure to a HIV-1 transactivator of transcription proteins led to ZO-1 downregulation, which was mediated by BMEC autophagy induction in vitro [264,265]. Furthermore, the transactivator of transcription was able to cross the BBB in a bidirectional manner [266]. Investigating the impact of this protein on the BBB is of special importance since it affects most cell types in the CNS, thereby contributing to neurotoxicity in HIV-1 associated neurocognitive disorders [267]. A recent review of Swingler et al. [268] highlights the applicability of the iPSC-derived organoids of different brain regions for various neurotropic viruses. By organoid modeling, the molecular regulation of neurotrophic viral infections, as well as cellular responses become more accessible. However, the known limitations of organoids, such as size and necrotic center formation, apply here as well.

### 5.3. Acute CNS Injuries

#### 5.3.1. Stroke

Functional and structural BBB alterations have been observed in vivo and in vitro in several stages of stroke. Due to the vast knowledge about the impact on the consequences of BBB breakdown in ischemic stroke, the current BBB in vitro models focus on replicating neuroinflammation through oxygen-glucose deprivation (OGD). The goal of these models is to restore the tightness of the BBB or to prevent its impairment, with the ultimate aim of facilitating drug development [269,270,271,272,273]. Cell death due to ischemia and peripheral immune cells causes more ROS, miRNAs, and damage-associated molecular patterns (DAMPs) to be generated, which activate glial cells. Microglial and astrocyte activation lead to pro-inflammatory mediator release (especially TNF-α and IL-1β), as well as the upregulation of pro-inflammatory genes in BMECs [12]. MMPs, which are primarily secreted by neutrophils, further digest TJ proteins [274]. Fattakhov et al. [269] and Kadir et al. [270] provided protocols for the establishment of triple-culture models in transwell setups. Lyu et al. [271] evaluated the restorative potential of stem cell therapies with their microfluidic NVU-on-a-chip. After the functional response to OGD, the authors tracked the infiltration of candidate stem cells through the BBB and observed that different types of stem cells exerted unique neurorestorative effects on the structural and functional integrity of the NVU rather than on the direct replacement of the neurons. A 3D microfluidic model by Cho et al. [147] with rat brain endothelial cells was developed to screen BBB-targeting drugs. The authors induced ischemic conditions by OGD, as well as showed the protective functions of the antioxidant edaravone and Rho kinase-inhibitor Y-27632. Wevers et al. [273] modeled stroke by introducing hypoglycemic conditions in a glucose-free medium, hypoxia by 10 µM of antimycin-A (an inhibitor of complex III of the electron transport chain), and disrupted the perfusion by moving the chips from a rocking platform to static conditions in the incubator. Due to the relatively high-throughput BBB chip, with 40 chips in parallel, this model was proposed as a tool for the drug screening of anti-inflammatory and free radical scavengers. Similar drugs were utilized in the NVU organoid model of Nzou et al. [272] when they were investigating their neuroprotective impacts. After inducing hypoxia by exposing the organoids to 0.1% O_2_ for 24 h, BBB breakdown, as well as pro-inflammatory cytokine and ROS production was observed.

#### 5.3.2. Traumatic Brain Injury

BBB impairment has been observed in all stages of traumatic brain injury (TBI), ranging from mild to severe. The disruption of the BBB occurs within the first hours after TBI due to neuroinflammation, and this may remain for years [275,276]. Despite decades of investigations through using in vivo models, there is no efficient neuroprotective treatment for TBI that has yet passed clinical trials. This is partly caused by the incomplete knowledge about molecular mechanisms in the complex pathophysiology of TBI. Therefore, in vitro models could be helpful in investigating the underlying structural and functional alterations at the cellular level. Different methods have been employed to induce neurotrauma in vitro since TBI itself can also result from different impacts, such as blunt-force, blast, or compression [277]. These injury models are induced statically or dynamically by mechanical or chemical forces. Static mechanical injuries are impact-based methods such as weight drop. Dynamic methods include microfluidic compartmentalization or scratch assays, as well as stretch-induced traumas that are caused by culture monolayer or axonal stretching. Chemical injuries are used for the microenvironment investigations of post-TBI tissues, and these include altering culture conditions with nutrient or oxygen deprivation, or treatments with chemicals [278]. Schlotterose et al. [279] developed a 3D-printed device for standard and non-standard cell culture applications to induce hydrostatic pressure on cells, resulting in their perturbation. As such, they were able to provoke TBI-typical hallmarks, such as cell death, decreased neuronal functionality, neural axon swelling, and a decrease in BBB tightness. Uni- and biaxial cell stretch models were set up by Rosas-Hernandez et al. [280,281] to assess cell viability and impact on the BBB after mild TBI. The model was based on rat BMECs on a silicone membrane, which was stretched utilizing a custom-made stretching device. They observed a deformation-dependent increase in cell death and apoptosis after high magnitudes of stretch, whereas the metabolic activity in the stretched BMECs was already decreased after a lower magnitude of deformation. Interestingly, low-magnitude stretching enhanced TJ protein expression, indicating its potentially protective role in BBB integrity. However, the authors also argued that experiments should be repeated with human BMECs since their results were shown to be contrary to the results of other studies that used mouse BMECs instead, thus indicating species-to-species differences. Salvador et al. [282] provided a protocol through which to perform a stretch- and OGD-induced in vitro model that could be used to mimic the impact of TBI on mouse BMECs. This approach can also be performed with human BMECs so as to serve a more physiologically relevant model. A systemic review by Wu et al. [283] summarized the current, at the time, in vitro models for TBI up until 2021. Only a very small proportion included investigations of the BBB alterations during trauma. Therefore, future studies are needed to reveal the impact of BBB dysregulation in the different forms of TBI. Furthermore, other cell types than solely BMECs, astrocytes, or neurons should be included in the setups. For example, pericytes were shown to rapidly detach from BMECs after TBI in a mouse model, but they were significantly enhanced again after 5 days, which indicates a biphasic pericyte regulation in acute TBI [284].

#### 5.3.3. Epilepsy

For decades, it has been discussed if BBB leakage is the cause or consequence for seizure development and aggravation [203,285]. A major problem in the treatment of epilepsy is the development of drug resistance in a third of patients. This treatment failure could be caused by an excessive upregulation of efflux transporters, such as P-gp or the molecular target that are changes caused by anti-epileptic drugs themselves [47,286]. To study the interactions of epileptic tissue and BBB, several in vitro models have been developed. However, although being labeled as in vitro models, many of these models have been of organotypic brain slices or whole brains that were isolated from rodents [287,288,289]. Seizure-like events can be induced by low magnesium concentrations, multiple doses of kainic acid, or 4-aminopyridine (a Ca^2+^ channel blocker) [290]. To achieve personalized drug development for patients, the organotypic cultures of tissue obtained from epileptic patients can be isolated and used in co-culture contact or in non-contact transwell models along with BMECs, as well as in brain tissues that exhibit seizure-like activities [291]. This allows one to display TJ and adhesion protein expressions in the monolayer, as well as to investigate the bidirectional interactions of BMECs with epileptic tissue, especially after the application of different anti-epileptic drugs [286]. So far, there are no in vitro models of epilepsy that incorporate more BMECs and astrocytes/isolated brain tissue [200]. Furthermore, working with isolated human brain tissue is limited due to high complexity and low availability [292]. Yamanaka et al. [293] reviewed the neuroinflammatory role of pericytes and their important impact in epilepsy pathogenesis. Pericytes undergo redistribution and remodeling during epileptic events, thereby causing BBB alterations due to responses to pro-inflammatory cytokines, as well as pericyte-glial scarring at permeable microvessels. Therefore, including pericytes in epilepsy in vitro co-culture models would possibly provide more physiologically reliable results for human drug discovery [122,294].

### 5.4. CNS Tumors

Brain cancers display significant heterogeneity, arising either from the CNS itself or due to spreading from the periphery as metastases through a permeable BBB. The BBB experiences several alterations during tumor cell invasion. This is also often referred to as the blood–brain tumor barrier, as microvessels have highly heterogeneous permeability following the active efflux of molecules, enhanced angiogenesis, and inflammation when they are combined with less blood flow due to tumor growth. The highest permeability is frequently found at lesion cores [200]. Novel-formed microvessels within the tumor tissue exhibit neither sufficient intercellular junctions nor transport systems [47]. With tumor progression, astrocytes lose their connection to the vasculature and interact with tumor cells, thereby regulating proliferation, immune cell invasion, and drug responses [295]. Furthermore, edemas due to AQP-4 upregulation is regularly observed in brain tumors [12]. Otherwise, BBB permeability in brain tumors is a two-edged sword. Although the barrier is leaky, chemotherapeutics either cannot enter, because the breakdown of the BBB is unequally distributed, or they are removed again from the CNS by active efflux mechanisms [296]. Researchers have already attempted to exploit neuroinflammation in order to open the BBB. Blethen et al. [296] provide an overview of methods, such as low-intensity-focused ultrasound, which can be employed to modulate the blood–brain tumor barrier in order to enhance drug uptake. These methods induce a secondary inflammation process that leads to the transient intercellular opening of BMECs and is mediated by the elevated release of DAMPs, TNF-α, IL-8, and heat shock proteins into the brain parenchyma. Although this approach was already proven to be successful in some preclinical studies, there is more research needed for the fine-tuning of low-intensity-focused ultrasound. Tumor modeling benefits greatly from 3D in vitro modeling as the tumor microenvironment can be mimicked for chemotherapeutic screening and metastasis studies. Experimental setups include multiple cell types in 3D scaffolds, thereby adding tumor spheroids in the brain parenchyma or vessel compartment [297,298,299,300]. Seo et al. [301] established a glioblastoma multiforme spheroid, which they placed in a vascularized 3D hydrogel to study chemosensitivity and the drug delivery associated with the BBB in this tumor. They added TNF-α and saw enhanced BBB permeability and monocytic THP-1 cell adhesion to BMECs due to higher adhesion molecule expression. The authors also highlighted the importance of other NVU cell types, such as astrocytes for BBB integrity, but also their prohibitive role in cancer metastasis. A recent study by Zhang et al. [302] suggested the enhanced adhesion of circulating breast cancer cells with metastasis potential to the brain endothelium. The microfluidic model included varying shear stress, selecting those cells with the highest adhesion to BMECs. As such, the authors observed not only elevated brain metastasis gene expressions in circulating breast cancer cells, but also increased transmigrations through the BBB in a transwell model. Moreover, their adhesion and proliferation within polyacrylamide gels (0.6 kPa) (which were coated with collagen I) was better than the one of wild-type cancer cells, which the research group associated with a better survival in the soft brain microenvironment. Finally, immunocompetent glioblastoma organoids could bring about a breakthrough in personalized medicine when co-cultured with patient blood/tumor-derived immune cells for the development of immunotherapies [303]. One patient-specific glioblastoma model was established by Cui et al. [304], who aimed to optimize patient-specific responses to programmed cell death protein-1 checkpoint immunotherapy by dissecting the immunosuppressive tumor microenvironment heterogeneity.

### 5.5. Neurodegeneration

#### 5.5.1. Alzheimer’s Disease

Independently of whether it is the cause or consequence, the involvement of BBB disruption in AD has been proven to a large extent [305]. Evidence for the BBB breakdown in AD patients also comes from plasma proteins like fibrin(ogen), thrombin, albumin, and the immunoglobulins that are found co-localized with Aβ plaques [16,306,307,308]. In AD, glial cells are primarily engaged in neuroinflammation. Studies have shown the activated astrocytes and microglia around plaques, which release pro-inflammatory cytokines and trigger further inflammatory processes. Astrocyte alterations besides astrogliosis in AD comprise an impairment in glucose metabolism, a decreased expression of glutamate transporters, and an imbalance in potassium [309]. A recent small nuclear RNA study by Xu et al. [310] of human postmortem brains revealed the transcriptomic changes in astrocytes and microglia in pathogenic conditions. Although astrocytic changes had common features in AD and PD, the regional differences were provocative toward one of the pathologies. In contrast, the microglia showed unique disorder gene transcriptomic patterns. In vitro BBB models of AD incorporate inflammatory regulation by adding pro-inflammatory cytokines and Aβ. Spampinato et al. [305] investigated the peripheral blood mononuclear cell transmigration across the barrier under Aβ exposure and pro-inflammatory conditions, as well as provided a protocol for a transwell model setup. Schreiner et al. [311] summarized the current AD BBB in vitro models in great detail.

#### 5.5.2. Amyotrophic Lateral Sclerosis

Blood-derived components like the thrombin, immunoglobulin G, or hemoglobin that have been found in ALS patient postmortem brains indicate BBB breakdown. Furthermore, a decrease in TJ protein expression, astrogliosis with detached astrocyte endfeet, and severe pericyte degradation have also been observed [16,312]. In ALS progression, adaptive immunity may contribute to the disease severity, and recent studies have highlighted the CNS infiltration of T cells, as well as an overall shift toward pro-inflammatory T cell subsets and reduced CD4+ T cells [313]. For example, CD4+ Tregs are reduced during disease progression. Their suppressive capability is also diminished in vitro by isolating patient T cells. Interestingly, removing Tregs from their environment could restore this characteristic feature, thererby displaying a possible therapeutic target for the autologous transplantation of T cells [314]. Most studies regarding BBB breakdown in ALS have been performed in *SOD1* mutant rodent models or in postmortem human tissues [315,316]. In vitro, not many models have been established until recently. The majority of models have been of the cell cultures of motor neuron cell lines with the abovementioned mutations, or of patient-derived motor neurons. Therefore, the focus has been to understand cellular mechanisms more specifically. In recent years, there has been a significant focus on using iPSCs to better understand the causes of ALS and to develop effective treatment approaches [317]. A recent review by Arjmand et al. [318] provides an input about how an ALS organ-on-a-chip can be approached. Embryonic or neural stem cells or iPSCs can be derived from patient fibroblasts. The microenvironment should involve an excess of glutamic acid to mimic the glutamate-induced excitotoxicity.

#### 5.5.3. Parkinson’s Disease

α-Synuclein acts in a pro-inflammatory manner via different mechanisms in PD [81]. On the one hand, it binds to microglia and astrocytes via toll-like receptor-2 and -4, thus leading to the release of inflammatory mediators [319,320]. On the other hand, monomeric α-synuclein was shown to induce the release of pro-inflammatory cytokines/chemokines by pericytes in a transwell model, thus leading to reduced BMEC integrity [321]. Enhanced BBB impairment was also seen in PD patients through using quantitative MRI imaging [322]. However, although pre-formed α-synuclein fibrils lead to a downregulation of TJ proteins, BBB breakdown seems to be dependent on multiple inflammatory events since the functionality of the BBB by the addition of α-synuclein alone was not altered significantly in vitro [323]. Therefore, including multiple cell types to model PD is crucial for reflecting the pathology, and for performing reliable experiments. Pediaditakis et al. [324] established a “human Substantia Nigra Brain-Chip” consisting of a PDMS chip with two perfusable microchannels and an ECM component mixture-coated PDMS membrane in between. Thus, they introduced iPSC-derived dopaminergic neurons, human primary astrocytes, pericytes, and microglia into one of the two channels, and iPSC-derived BMECS on the surface of the opposite channel. Continuous exposure of α-synuclein fibrils in the “brain” channel showed an accumulation of pSer129-αSyn fibrils, mitochondrial impairment, and an enhanced BBB permeability after a few days.

#### 5.5.4. Huntington’s Disease

A study by Vignone et al. [109] compared the iBMECs of HD patients with those of the unaffected controls. They aimed to investigate whether changes in the transcriptomic profiles indicated if the BMECs themselves were functionally compromised when promoting BBB dysfunction. The results indicated alterations in the BBB properties, and in functions such as receptor-mediated transcytosis. A gene expression analysis of iBMECs by Linville et al. [325] showed similar results regarding TJ proteins, while they could not observe the impaired paracellular permeability in their transwell setup. Furthermore, the authors saw differences in their immune cell adhesion and immune activation transcripts between the juvenile HD iBMECs and adult postmortem HD BMECs. Therefore, they suggested BBB gene expression changes due to the elevated CAG repeat expansion during HD progression. Besides these studies using iBMEC monocultures, not many investigations have been conducted on the differentiation of other HD-relevant cell types. However, this could be due to the difficulty of replicating differentiation protocols (for instance, striatal medium spiny neurons [326]). There are still many tasks that need to be completed in order to accurately model the BBB for HD. One key area that needs development is the creation of an in vitro co-culture model that includes other CNS resident cells to better determine the BBB phenotype.

## 6. Discussion

The barrier function of the BBB is strictly regulated by various cell types in the CNS. While in the past BMECs have primarily been employed for in vitro modeling, investigations on the surrounding astrocytes, pericytes, microglia, neurons, and oligodendrocytes have shed new light on the importance of these CNS-resident cells. In neuroinflammation, both central and peripheral cells produce pro-inflammatory mediators that lead to BBB impairment. Inflammation-inducing factors vary across the spectrum of neurological diseases, such as the production of autoantibodies, the release of ROS and DAMPS due to ischemia and trauma, and the influence of metastases on the microvascular efflux pumps and permeability. While a huge variety of different models exist for some neuropathologies, there is a devastating lack of in vitro models for others. Some disease models could benefit from the already-established setups of other diseases. While monoculture transwells may not be suitable for more complex physiological research questions or personalized approaches, they still offer advantages in conducting basic high-throughput toxicity or permeability investigations. Models that incorporate multiple cell types better mimic human (patho)physiology, even more so when applied to a 3D model setup with an ECM substitute and shear stress. The tunability of the incorporated cells can be established by using patient-derived iPSCs to enable more personalized research regarding drug tolerance, toxicity, and efficacy. It is worth noting that as the model becomes more complex, there are more potential sources of error that can accumulate. To validate the model, the cell-type identity and viability need to be confirmed. Permeability can be measured by TEER, or via size-defined particles that can pass a leaky barrier. Appropriate barrier formation is evaluated via TJ protein detection. For the future, a robust and reliable platform mimicking different neurological diseases that are as close to in vivo conditions as possible is desirable in order to reduce lab-to-lab aberrancies. Although in vitro models will never fully reflect human physiology, they are great tools that allow for increasing reduction and replacement of animal models.

## 7. Conclusions

There are several key considerations for BBB in vitro modeling that have been addressed in this review. First, considerations need to be made regarding the “why” of the experiment. The model setup can completely differ depending on whether the transport of substances across a BMEC monolayer is examined, or how the tumor microenvironment in glioblastoma organoids changes due to different immunotherapy approaches. Each pathology brings different characteristics and biological requirements that need to be adapted accordingly in the BBB model. For example, the majority of patients with NMOSD exhibit AQP-4 antibody seropositivity. To investigate the pathomechanism of these antibodies against astrocytes, it is essential to evaluate the AQP-4 expression of these cells before conducting experiments. Investigations of AD or PD need the presence of Aβ or α-synuclein fibrils, respectively, on the “brain” side. Acute CNS injuries need to be induced before the investigations, such as OGD in ischemia models or those in the differently provoked injuries in TBI. Additionally, when modeling neuroinflammation, it is important to choose a method for inducing inflammation. Most models aim for stimulation with LPS or TNF-α, but the addition of other pro-inflammatory substances can be considered as well. Moreover, the cell and ECM types, as well as the general setup (simple BBB vs. full NVU), need to be considered. For experiments on the function of the different transporters or receptors of BMECs, a BMEC monoculture model that shows the respective protein expression can be sufficient. However, investigations of the different cell types’ interactions might require more complex setups. For transwell models, the multi-well format, pore size, and membrane material, as well as a BM substitute need to be defined. For hydrogel-based models, the hydrogel composition and cross-linking substance need to be evaluated; in addition, some additives might be required for enhanced cell viability. Organoids need a cocktail of growth factors and an ECM for proper differentiation. Furthermore, the time frame of each experimental approach may vary. Utilizing iPSCs for the model can be time-consuming since the differentiation protocols require several weeks to complete. Then, there is a short time window where the differentiated cells can be used for the model before de-differentiation. Primary cells can be commercially purchased or isolated directly (if ethically applicable), but they can only be used for a few passages, which limits the number of attempts and cells for the model establishment. In contrast, immortalized cell lines can be used over many passages in higher quantities. Moreover, in particular for drug safety studies in pharmaceutical production, a high-throughput platform is essential. Often, the quantity of the model compromises its quality or, in this case, its complexity. Hence, high-throughput models most likely will not require time-intensive fabrication. For some models, additional devices are needed. For the introduction of shear stress, a perfusion system is required. This can involve different pumping systems or an incubator-safe rocking platform. BBB/NVU-on-a-chip models need specially fabricated chips made of PDMS or similar materials. It should be noted that there are various printing techniques used for 3D bioprinted models that require different devices based on the printing material and complexity of the model. To measure the TEER, specialized TEER meters need to be purchased. Immunofluorescence staining requires fluorescence microscopes. If experiments are conducted in 3D setups, a laser confocal microscope can be beneficial for creating a 3D representation of the staining. Finally, the culture time is dependent on the cells used and the experimental procedure performed. Experiments are usually started when BMECs have been cultured long enough to develop a tight monolayer. The formation of TJs and low permeability need to be assessed by TJ marker immunofluorescence staining, TEER measurements, or permeability assays. The enhancement of TJs can be reached by the addition of glucocorticoids, but this should be avoided in inflammation-related studies. Additional incubation time might be necessary, if inflammation needs to be provoked or if the effects of different drugs, cytokines, ROS, antibodies, or immune cells on the BBB are being studied.

Ultimately, the selection of a suitable model setup and its complexity depend on the specific research question at hand. As the British statistician George E. P. Box once said (which is also applicable for biological models): “All models are wrong, but some are useful” [327]. 

## Figures and Tables

**Figure 1 ijms-24-12699-f001:**
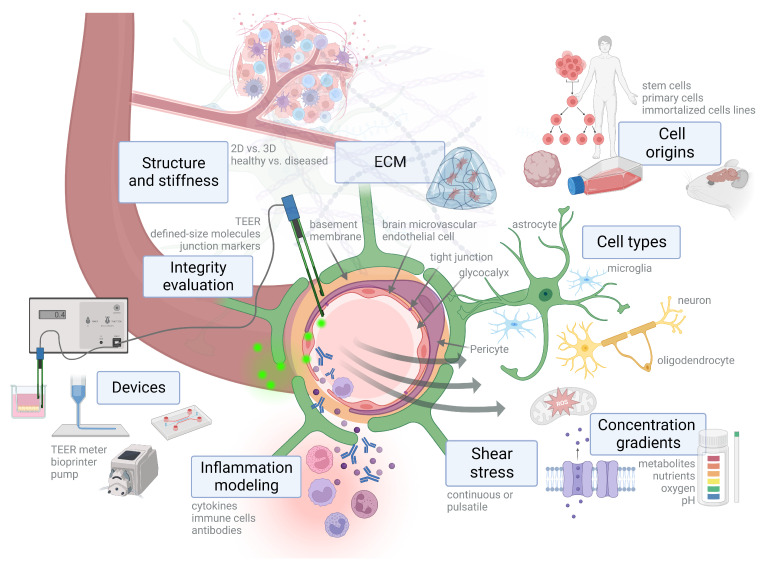
Considerations for BBB/NVU model development. Adapted from Cameron et al. [87]. Created with BioRender.com.

**Table 1 ijms-24-12699-t001:** Key features of the neurological diseases with BBB impairment.

Disease	Key Features	References
**CNS Autoimmune Diseases**	Complex interplay of genetic and environmental factors Disease onset and severity are driven by triggers, such as infections or tumors Autoantibodies and reactive T cells	[189,190]
Multiple Sclerosis	Chronic inflammatory demyelinating disease with autoimmune aspects. Th1 and Th17 CD4+ T cells might be autoreactive, thus producing interferon-γ (by Th1 cells), IL-17, and IL-22 (by Th17 cells), which dysregulate TJ proteins and upregulate adhesion molecules Human leukocyte antigen gene polymorphisms and viral infections (Epstein-Barr Virus) Inflammatory processes are provoked by neurotoxic astrocytes and microglia, which attract immune cells and lead to progressive or relapsing axonal demyelination with oligodendrocyte apoptosis, astrogliosis, axonal loss, and secondary neurodegeneration	[16,191,192,193]
NMDARE	NMDA receptor autoantibodies lead to an internalization of the receptor Autoimmune Encephalitis with psychiatric symptoms, autonomic fluctuations, and seizures	[190,194]
NMOSD	The majority of patients are seropositive for the antibodies that are against AQP-4 on the endfeet of astrocytes that induce complement-dependent cytotoxicity Association with acute optic neuritis, myelitis, area postrema syndrome, and distinct MRI lesions	[189,195,196]
MOGAD	Patients have autoantibodies against MOG on the oligodendrocytes that can activate FcR-mediated antibody-dependent cellular cytotoxicity and which complement-dependent cytotoxicity Association with acute disseminated encephalomyelitis, optic neuritis, transverse myelitis, brain stem syndrome, and encephalitis	[189,197]
**CNS Infections**		
Bacterial Infections	The endotoxin LPS of gram-negative and lipoteichoic acid from gram-positive bacteria leads to BBB disruption via toll-like receptor-4, thereby inducing pro-inflammatory cytokines for the innate immune response The endotoxin hypothesis suggests that gut-bacterial endotoxin causes/contributes to neurodegeneration	[198,199]
Viral Infections	Entry via transcytosis, and the infection of BMECs or CNS-invading immune cells occurs as a Trojan horse HIV-1, SARS-CoV-2, and others reduce TJs and upregulate adhesion molecules/cytokines Can cause/trigger other neurological disorders, such as dementia, epilepsy, autoimmune diseases, etc.	[200,201]
**Acute CNS Injuries**		
Stroke	Ischemia (in 85% of cases) with nutrient and oxygen deprivation in the hyperacute phase (<6 h), whereby the TJ opening occurs due to hypoxia-induced ROS and transcription factors (hypoxia-inducible factor-1α). This is followed by cytotoxic (ionic) edema due to the cell swelling that occurs because of ion and water homeostasis imbalance A refractory period with increasing BBB openings (hours or days) leads to plasma protein entry in the CNS, possibly resulting in a vasogenic edema by fluid increase due to BBB openings and the extravasation of plasma proteins Potential hemorrhagic conversion with a loss of microvessel integrity and the entry of blood components into the CNS	[47,81,202,203,204]
Traumatic Brain Injury	Inflammation by microglia and astrocyte activation (via albumin/DAMPs) and by neutrophilic infiltration; chronic inflammation might be linked to enhanced white matter-/neurodegeneration and encephalopathy Elevated neuronal stimulation and toxicity through the overproduction of glutamate receptors and GABA-receptor internalization Further BBB breakdown and dysregulated metabolism by ROS and other pro-inflammatory molecules, as well as in the sub-acute stages of TBI that is enhanced by NOX and activated by MMPs	[75,203,205,206]
Epilepsy	Recurrent seizures whose frequency has been shown to correlate positively with BBB breakdown Higher K+ influx to the CNS due to leukocyte–BMEC interactions. Astrocytes and microglia produce pro-inflammatory mediators that lower TJ protein expression and BBB tightness. The extravasation of serum proteins and immune cells augment the inflammatory responses The downregulation of the K+ channel subunit 4.1 (via albumin binding to TGF-β receptor) and excitatory amino acid transporter-2 (via TNF-α) in astrocytes cause an increase in the K+ and glutamate that enhance neuronal hyperactivity	[203,207]
**CNS Tumors**		
Primary Tumors	Glioblastoma multiforme (from astrocytes) is the most aggressive one, but does not metastasize Cancer cells release exosomes with VEGF-A and pro-inflammatory cytokines, which lead to a TJ decrease	[12,208]
Brain Metastases	A total of 30% of brain cancers, often derived from lung cancers, breast cancers, or melanomas Microvessels have heterogeneous permeability with their active efflux of molecules, enhanced angiogenesis, and inflammation	[152,203]
**Neurodegeneration**		
Alzheimer’s Disease	The most common cause of progressive cognitive impairment worldwide (increasing with the rising percentage of aged people) that has genetic and environmental risk factors Genetic forms (in total 1%) are caused by mutations in the genes *APP*, *PSEN1*, or *PSEN2* Neuroinflammation and degeneration, Aβ plaques, and neurofibrillary tangles (hyperphosphorylated tau)	[16,209]
Amyotrophic Lateral Sclerosis	Muscle weakness and atrophy with a progressive decrease in motor functions through motor neuron degeneration The 43 insoluble TAR-DNA-binding protein aggregations in the soma of motor neurons exhibit neurotoxic properties. This also occurs in other misfolded protein aggregates (with mutations in *SOD1* or *C9orf72*) in familial forms (10%) Glutamate-mediated excitotoxicity, increased oxidative stress, mitochondrial dysfunction, and neuroinflammation	[210,211,212]
Parkinson’s Disease	Age-associated neurodegenerative disease with dopaminergic neuronal loss in the substantia nigra Neuroinflammation mediated by Lewy body depositions (the aggregations of α-synuclein oligomers or fibrils) Genetic factors, environmental toxins, and mitochondrial dysfunction play a role	[213,214,215,216]
Huntington’s Disease	Glutamine (CAG) repeat expansion mutations in the *HTT* gene Affected areas are the basal ganglia (i.e., the dorsal striatum and cortical compartments) Reactive microglia and astrocytes play major roles in neuroinflammation generation	[108,217,218,219]

**Abbreviations:** A*β*, amyloid-beta; BBB, blood–brain barrier; CNS, central nervous system; DAMPs, damage-associated molecular patterns; GABA, gamma-aminobutyric acid; HIV-1, human immunodeficiency virus-1; IL, interleukin; LPS, lipopolysaccharide; MOG, myelin oligodendrocyte glycoprotein; MOGAD, MOG-associated antibody disease; NMDARE, N-methyl-D-aspartate receptor encephalitis; NMOSD, neuromyelitis optica spectrum disorders; NOX, nicotinamide adenine dinucleotide phosphate oxidase; ROS, reactive oxygen species; TGF-*β*, transforming growth factor-beta; Th, T helper; TJ, tight junction; TNF-*α*, tumor necrosis factor-alpha; SARS-CoV-2, severe acute respiratory syndrome coronavirus type 2; and VEGF-A, vascular endothelial growth factor-A.

## Data Availability

Not applicable.

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
