# Peer review of "Blood–Brain Barrier Breakdown in Neuroinflammation: Current In Vitro Models"

_ijms, 2023, doi:10.3390/ijms241612699_

Round 1

Reviewer 1 Report

The article has an interesting approach that of inflammation-related BBB breakdown as studied in in vitro models. However, the review is disappointing in several ways. There are many restatements lifted from other reviews that are wrong, inaccurate, or misstated.  There are a lot of ambiguous or confusing sentences.  The choice of subtopics and literature seems somewhat arbitrary, especially from the supporting/confirming in vivo data.  It seems strange that lipopolysaccharide is not discussed in an article such as this.  Although a long article, specific sections seem to have been treated superficially, especially those section related to specific diseases.  Nevertheless, the later parts of review are well done, such as comparing the different models and matrix components. Specific comments are:

Ln 13: “Usually, 98 % of drugs…fail to enter the central nervous system (CNS)…”.  Please provide a reference that supports this statement. This often stated comment seems to have no basis and the actual number appears closer to 30%.

Ln 26-7:  “The BBB comprises only brain microvascular endothelial cells (BMECs) with the adjacent macroglia: Pericytes in the basement membrane (BM) and astrocyte endfeet enwrapping the capillaries. “  What constitutes the fundamental vascular BBB can be variably defined, especially now: it varies from just the BEC with astrocytes and pericytes and other cells signaling the formation of barrier features, but not participating in the physical barrier to inclusion of the basement membranes and glycocalyx.  The authors should reimagine this sentence.

Ln36-7: “…fenestrae as found in other tissues, they are connected via gap-, adherens- and tight junctions”. Confusing sentence because as written it implies fenestrae connect the endothelial cells to each other in peripheral capillary beds.  

Ln 52-3: “Only small lipophilic molecules with a molecular weight below 400 Dalton and less than eight hydrogen bonds can diffuse freely across the BBB.”  This sort of statement needs referencing.  It is also not true.  Many substances larger than 400 Da cross the BBB readily by lipid solubility 1-4.  Additionally, for reasons that are unclear, the Rule of 5 does not seem to apply to certain classes of substances, including anti-helminthics and peptides.

Ln 56-7: “All of them have a 56 specific direction …”. What does that mean? Glucose is facilitated diffusion and so bidirectional and others are unidirectional. 

Ln64: “Hence, most proteins, drugs…” Please justify with reference that MOST drugs do not enter the BBB because of efflux transporters.

Ln 67: If you discuss the BM, you should discuss the glycocalyx.

Ln 110: “The role of neurons in BBB integrity has become increasingly important recently. “  Ambiguous sentence: Do you really mean to say that neurons have suddenly evolved to be more important (literally what this sentence says) or do you mean our understanding of the role of the neuron is what has evolved? 

Ln 125: “Oligodendrocytes belong to the macroglia, but are not accounted for in the NVU, as  they do not contribute directly to BBB formation.”  NVU cells are generally taken to be those that influence BBB functions, not must formation. Additionally, oligodendrocytes do affect BBB function5.

Ln 183-4: “…cytokines are primarily secreted…central immune cells.” Not sure what the authors include as central immune cells: peripheral immune cells that have crossed the BBB? Microglia?  Other cells? Barrier cells (brain endothelial cells, choroid plexus epithelium), astrocytes, and pericytes also secrete cytokines.

Ln 187-9: These sentences are ambiguous and can be read that it is the cytokines within the CNS that result in disruption. But it is the circulating cytokines that are more clearly known to do this.

Ln 228-30: “By binding to…BMECs, IL-6 promotes their secretion of …TNF-α .”  Nearly every paper that has deliberately looked for TNF secretion by BMECs has failed to find it.

Ln 280: “Chemokines” are a subgroup of Cytokines, so to have them as a subheading equal to Cytokines (3.2 vs 3.3) is confusing.

Ln 296: What is meant by CXCL12 “loses its polarization”?  If it translocates selectively to the luminal membrane, is not this polarization?

Ln 423: “these cultures can only be utilized for basic research questions like toxicity and proliferation evaluations”. Why? On the one hand, the limitations they have might also make them unreliable for these kind of experiments as well. On the other hand, once shown they possess the relevant characteristics, why can’t they be used for transport experiments, characterization of secretion or immune response, etc? 

Ln 718: Remember to formally define T as well.

LN 739:  Sucrose and caffeine are very different markers as caffeine is a substrate for the adenosine transporter. Actually, Kadry et al does not mention caffeine in his MS

Ln 739-41: “…small molecules…are more suitable for subtle BBB dysfunction, while markers like fluorescence-labeled dextrans can  be utilized for BBB integrity or permeability evaluations. “  This statement makes no sense because as used here, these are all terms related to BBB disruption.  Different markers and probes can indeed be used in special ways, but this sentence does not help with understanding how. 

Table 1. NMDARE, NMOSD, MOGAD all need to be defined. 

Ln 759:  Immune cells do not cross the BBB in MS because of “breakdown” or leakage (if that were the mechanism, red blood cells would enter the CNS as well), but because of enhanced diapedesis. So “dysfunction” would be a better word here.

Ln844:  SARS-CoV-2 hard to keep track on. I am sympathetic to that, but surely you need to review the article by Rhea6given its direct relation to the topic of BBB and its prominent journal. 

Ln851: “Kase et al. observed that the major SARS-CoV-2 strain spike proteins were able to  infect microglia,…” I believe they misquote Kase et al. As written, Kase would have only studied the spike proteins, but they used viral particles to study productive infection.

Ln 1044-1048:  Two sentences for Neurodegeneration just highlights that you have decided not to review the topic in this way. Also, you elsewhere review neurodegenerative diseases such as AD and Parkinson’s. I would suggest you delete this two sentence section.

Other grammatically incorrect or confusing sentences & Typos:

Ln 136-7: Both brain resident cells, such as glial cells, and peripheral immune cells that con tribute to neuroinflammation. 

Ln 160: “This A1 type astrocyte type…” 

Ln347: “messes” is vernacular.

Ln 381: “…which makes it is crucial…”

Ln 386: “entity”, do you mean “characteristics”?

Ln 702-4: “However, those methods do not reflect the dead cell compartment reductions in a 3D ECM occur only slowly [193]. “

Many ambiguous or grammatically incorrect sentences.  This will take some effort to correct.

Author Response

The article has an interesting approach that of inflammation-related BBB breakdown as studied in in vitro models. However, the review is disappointing in several ways. There are many restatements lifted from other reviews that are wrong, inaccurate, or misstated. There are a lot of ambiguous or confusing sentences. The choice of subtopics and literature seems somewhat arbitrary, especially from the supporting/confirming in vivo data. It seems strange that lipopolysaccharide is not discussed in an article such as this. Although a long article, specific sections seem to have been treated superficially, especially those section related to specific diseases. Nevertheless, the later parts of review are well done, such as comparing the different models and matrix components.

Response: We would like to thank the reviewer for his critical and thoughtful review. We regret that some of the restatements lifted from other reviews were wrong, inaccurate, or misstated and have now corrected them (see our response to specific comments below). We were also working on improving the English language of our review article by correcting ambiguous or grammatically incorrect sentences. We also tried, as recommended in your comments, to improve the choice of subtopics and literature. However, since the focus of our review article is on in vitro models we decided not to focus on in vivo data. Finally, we have now also included an in vitro model using lipopolysaccharide as recommended.

Ln 877-887 “Brown et al. investigated the impact inflammatory stimulation on the BBB induced by LPS (mimicking systemic bacterial infection) or pro-inflammatory cytokines (which could result of a local or systemic inflammation) in their 3D microfluidic NVU dual chamber model. By applying either 100 µg/mL of LPS or a cytokine mix with 100 ng/mL (IL-1, TNF-α and MCP-1,2) to the vascular compartment, inflammation was induced. They then looked into the metabolic profile on both sides over the time of 24 h of exposure. Interestingly, they found pro-inflammatory cytokine activation in both vascular and brain compartments, whereas at later time points this activation was only seen in a subset in both compartments. This suggests the activation of pro-repair cytokines at later time points within the brain section, as the vascular compartment remains more pro-inflammatory [76].”

Reference:

  1. Brown, J.A.; Codreanu, S.G.; Shi, M.; Sherrod, S.D.; Markov, D.A.; Neely, M.D.; Britt, C.M.; Hoilett, O.S.; Reiserer, R.S.; Samson, P.C.; et al. Metabolic consequences of inflammatory disruption of the blood-brain barrier in an organ-on-chip model of the human neurovascular unit. Journal of neuroinflammation 2016, 13, 306. https://doi.org/10.1186/s12974-016-0760-y.

The section related to specific diseases was meant as an introduction to the last part of our review (in vitro model) which may explain the superficial treatment of specific diseases.

Finally, we would like to thank you for your positive feedback on the later parts of our article.

Response to specific comments:

Point 1: Ln 13: “Usually, 98 % of drugs…fail to enter the central nervous system (CNS)…”. Please provide a reference that supports this statement. This often stated comment seems to have no basis and the actual number appears closer to 30%.

Response 1: We regret that we missed to provide a reference supporting our statement. According to Pardridge Drug Disc Today 2007, 12: 54-61 “Essentially 100% of large-molecule drugs and >98% of small-molecule drugs do not cross the BBB”. This paper is highly cited, also by very recent publications. However, after your statement we are aware that this issue is discussed controversially. Therefore, we have amended this statement and added the following reference: Banks WA and Greig NH, Future Med. Chem. (2019) 11(6), 489–493):

Ln 13-17 “Usually, 98% of small-molecule drugs, and close to 100% of large-molecule drugs most other substances, fail to enter the central nervous system (CNS) through the tight barrier of endothelial cells [1]. Of those drugs that are specifically developed for the treatment of CNS diseases, approximately 30% fail to access the CNS across the endothelial cell layer [2].”

References:

  1. Pardridge, W.M. Blood-brain barrier delivery. Drug discovery today 2007, 12, 54–61. https://doi.org/10.1016/j.drudis.2006.10.013.
  2. Banks, W.A.; Greig, N.H. Small molecules as central nervous system therapeutics: old challenges, new directions, and a 1226 philosophic divide. Future medicinal chemistry 2019, 11, 489–493. https://doi.org/10.4155/fmc-2018-0436.

Point 2: Ln 26-7: “The BBB comprises only brain microvascular endothelial cells (BMECs) with the adjacent macroglia: Pericytes in the basement membrane (BM) and astrocyte endfeet enwrapping the capillaries.“ What constitutes the fundamental vascular BBB can be variably defined, especially now: it varies from just the BEC with astrocytes and pericytes and other cells signaling the formation of barrier features, but not participating in the physical barrier to inclusion of the basement membranes and glycocalyx. The authors should reimagine this sentence.

Response 2: We have now rephrased this sentence as suggested:

Ln 29-33 “There are different definitions of the vascular BBB: It varies from just the brain microvascular endothelial cells (BMECs) with pericytes in the basement membrane (BM), astrocyte endfeet enwrapping the capillaries and other cells signaling the formation of barrier features, but not themselves participating in the physical barrier, to the inclusion of the BM and glycocalyx.”

Point 3: Ln36-7: “…fenestrae as found in other tissues, they are connected via gap-, adherens- and tight junctions”. Confusing sentence because as written it implies fenestrae connect the endothelial cells to each other in peripheral capillary beds.

Response 3: The sentence is now rephrased for better understanding. Fenestrae are small pores between endothelial cells in capillaries of other tissues and are not connecting cells with each other. Ln 42-45 “In other tissues, there are small pores, so-called fenestrae, between the endothelial cells of the capillaries. In contrast, BMECs exhibit a tight layer with the help of gap-, adherens- and tight junctions (TJs) that bring them into proximity [4,5].”

References:

  1. Abbott, N.J.; Patabendige, A.A.K.; Dolman, D.E.M.; Yusof, S.R.; Begley, D.J. Structure and function of the blood-brain barrier. Neurobiology of disease 2010, 37, 13–25. https://doi.org/10.1016/j.nbd.2009.07.030.
  2. Stamatovic, S.M.; Johnson, A.M.; Keep, R.F.; Andjelkovic, A.V. Junctional proteins of the blood-brain barrier: New insights into function and dysfunction. Tissue barriers 2016, 4, e1154641. https://doi.org/10.1080/21688370.2016.1154641.

Point 4: Ln 52-3: “Only small lipophilic molecules with a molecular weight below 400 Dalton and less than eight hydrogen bonds can diffuse freely across the BBB.” This sort of statement needs referencing. It is also not true. Many substances larger than 400 Da cross the BBB readily by lipid solubility. Additionally, for reasons that are unclear, the Rule of 5 does not seem to apply to certain classes of substances, including anti-helminthics and peptides.

Response 4: The statement is referenced (Pardridge WM, Journal of Cerebral Blood Flow & Metabolism. 2012;32(11):1959-1972) and stated as such in many recent reviews. However, we are aware that there are different opinions and findings that deviate from this statement. Therefore, the statement was extended as follows, including the respective reference:

Ln 61-68 “Many reports from the literature state that only small lipophilic molecules with a molecular weight below 400-600 Da and few (<8) hydrogen bonds, as well as gases, can diffuse freely across the BBB. However, there is evidence that there is no distinct cutoff, but rather a molecular weight penalty. Indeed, there are certain molecules with higher molecular weight that have been proven to diffuse through the BBB by lipid solubility, the largest known being cytokine-induced neutrophil chemoattractant-1 with about 7 kDa. Furthermore, some substrate classes, such as anti-helminthics and (opiate) peptides and their analogs have the ability to cross the BBB by diffusion as well despite higher molecular weight [2,11,12].”

References:

  1. Banks, W.A.; Greig, N.H. Small molecules as central nervous system therapeutics: old challenges, new directions, and a philosophic divide. Future medicinal chemistry 2019, 11, 489–493. https://doi.org/10.4155/fmc-2018-0436.
  2. Pardridge, W.M. Drug transport across the blood-brain barrier. Journal of cerebral blood flow and metabolism : official journal of the International Society of Cerebral Blood Flow and Metabolism 2012, 32, 1959–1972. https://doi.org/10.1038/jcbfm.2012.126.
  3. Banks, W.A. Characteristics of compounds that cross the blood-brain barrier. BMC neurology 2009, 9 Suppl 1, S3. https://doi.org/10.1186/1471-2377-9-S1-S3.

Point 5: Ln 56-7: “All of them have a 56 specific direction …”. What does that mean? Glucose is facilitated diffusion and so bidirectional and others are unidirectional.

Response 5: After proofreading of the respective resources, we adapt the sentence according to the reviewer’s suggestion:

Ln 73 “Many of them have a specific direction and substrate precision.”

Point 6:  Ln64: “Hence, most proteins, drugs…” Please justify with reference that MOST drugs do not enter the BBB because of efflux transporters.

Response 6: We changed the word “most” to “these”, as the reviewer pointed out the incorrect use of “most” here (Ln 81).

Point 7: Ln 67: If you discuss the BM, you should discuss the glycocalyx.

Response 7: A description about the glycocalyx was added below the passage about the BM.

Ln 90-102 “Towards the vascular lumen, BMECs express the glycocalyx, which is a thin layer of villiform substance [18]. Its major components are proteoglycan protein polymers and glycosaminoglycan chains, including herparan sulfate, chondroitin sulfate and hyaloronic acid (HA), and their associated binding proteins [19]. The glycocalyx is important for many physiological functions of the BBB. Among them, it maintains the low permeability of the BBB, prevents inflammation triggers and coagulation response [18]. Furthermore, it was shown to sense changes in the shear force of the blood flow, subsequently inducing the release of endogenous vasoactive mediators [20]. As it is negatively charged due to its composition, the glycocalyx forms an electrostatic barrier for negatively charged molecules, proteins, and plasma cells [21]. During inflammation, the glycocalyx sheds off the BMECs to enable leukocyte binding to vascular cell adhesion molecules [18]. Alterations in the glycocalyx have been shown to affect the BBB integrity, which could promote the development of a broad range of 92 neurological diseases [22].”

The references are:

  1. Tietz, S.; Engelhardt, B. Brain barriers: Crosstalk between complex tight junctions and adherens junctions. The Journal of cell biology 2015, 209, 493–506. https://doi.org/10.1083/jcb.201412147.
  2. Jin, J.; Fang, F.; Gao, W.; Chen, H.; Wen, J.; Wen, X.; Chen, J. The Structure and Function of the Glycocalyx and Its Connection With Blood-Brain Barrier. Frontiers in cellular neuroscience 2021, 15, 739699. https://doi.org/10.3389/fncel.2021.739699.
  3. Salmon, A.H.J.; Satchell, S.C. Endothelial glycocalyx dysfunction in disease: albuminuria and increased microvascular permeability. The Journal of pathology 2012, 226, 562–574. https://doi.org/10.1002/path.3964.
  4. Lyu, N.; Du, Z.; Qiu, H.; Gao, P.; Yao, Q.; Xiong, K.; Tu, Q.; Li, X.; Chen, B.; Wang, M.; et al. Mimicking the Nitric Oxide-Releasing and Glycocalyx Functions of Endothelium on Vascular Stent Surfaces. Advanced science (Weinheim, Baden-Wurttemberg, Germany) 2020, 7, 2002330. https://doi.org/10.1002/advs.202002330.
  5. Zhao, F.; Zhong, L.; Luo, Y. Endothelial glycocalyx as an important factor in composition of blood-brain barrier. CNS neuroscience & therapeutics 2021, 27, 26–35. https://doi.org/10.1111/cns.13560.

Point 8: Ln 110: “The role of neurons in BBB integrity has become increasingly important recently.“ Ambiguous sentence: Do you really mean to say that neurons have suddenly evolved to be more important (literally what this sentence says) or do you mean our understanding of the role of the neuron is what has evolved?

Response 8: Of course, the reviewer’s point is valid and the sentence will be rephrased accordingly: Ln 140-141 “The understanding of the importance of neurons for BBB integrity has evolved over the last decades.”

Point 9: Ln 125: “Oligodendrocytes belong to the macroglia, but are not accounted for in the NVU, as they do not contribute directly to BBB formation.” NVU cells are generally taken to be those that influence BBB functions, not must formation. Additionally, oligodendrocytes do affect BBB function .

Response 9: The sentence was rephrased according to the reviewer’s suggestion:

Ln 156-157 “Oligodendrocytes belong to the macroglia and contribute as cells of the NVU to the BBB function.”

Point 10: Ln 183-4: “…cytokines are primarily secreted…central immune cells.” Not sure what the authors include as central immune cells: peripheral immune cells that have crossed the BBB? Microglia? Other cells? Barrier cells (brain endothelial cells, choroid plexus epithelium), astrocytes, and pericytes also secrete cytokines.

Response 10: The sentence was rephrased with the suggestions of the reviewer.

Ln 216-217 “In that state, cytokines are primarily secreted by peripheral immune cells that have crossed the BBB and brain resident cells that are capable of cytokine secretion.”

Point 11: Ln 187-9: These sentences are ambiguous and can be read that it is the cytokines within the CNS that result in disruption. But it is the circulating cytokines that are more clearly known to do this.

Response 11: We rephrased these sentences according to the reviewer’s statement to avoid confusion.

Ln 219-221 “Circulating pro-inflammatory cytokines provoke the disruption of the BBB, further inflammatory processes, neuron impairment and in the end neurological diseases.”

Point 12: Ln 228-30: “By binding to…BMECs, IL-6 promotes their secretion of …TNF-α.” Nearly every paper that has deliberately looked for TNF secretion by BMECs has failed to find it.

Response 12: We looked this up and have and now rephrased the sentence accordingly:

Ln 267-269 “In BMECs, this binding promotes the expression of adhesion molecules and the secretion of chemokines. The complex of IL-6 and soluble IL-6 is also able to activate astrocytes and microglia.”

Point 13: Ln 280: “Chemokines” are a subgroup of Cytokines, so to have them as a subheading equal to Cytokines (3.2 vs 3.3) is confusing.

Response 13: The reviewer has a valid point. The section of chemokines was assigned as subsection of cytokines and set into a correct order within the paper (3.2 Cytokines Ln 212, 3.2.4 Chemokines Ln280).

Point 14: Ln 296: What is meant by CXCL12 “loses its polarization”? If it translocates selectively to the luminal membrane, is not this polarization?

Response 14: This sentence was rephrased to avoid confusion:

Ln 297 “During neuroinflammation, CXCL12 loses its binding to CXCR4, relocates towards the microvessel lumen and thereby and enables lymphocyte and monocyte recruitment through the BBB.”

Point 15: Ln 423: “these cultures can only be utilized for basic research questions like toxicity and proliferation evaluations”. Why? On the one hand, the limitations they have might also make them unreliable for these kind of experiments as well. On the other hand, once shown they possess the relevant characteristics, why can’t they be used for transport experiments, characterization of secretion or immune response, etc?

Response 15: The reviewer has a valid point and the sentence was rephrased as proposed:

Ln 465-467 “These cultures are utilized in basic research for toxicity and proliferation evaluations, transport experiments, and characterization of secretion or immune response.”

Point 16: Ln 718: Remember to formally define T as well.

Response 16: As it is less confusing, T was replaced by „TEER”, which has been formally defined before.

Point 17: LN 739: Sucrose and caffeine are very different markers as caffeine is a substrate for the adenosine transporter. Actually, Kadry et al does not mention caffeine in his MS

Response 17: We regret this incorrect statement which has now been corrected accordingly.

Point 18: Ln 739-41: “…small molecules…are more suitable for subtle BBB dysfunction, while markers like fluorescence-labeled dextrans can be utilized for BBB integrity or permeability evaluations.“ This statement makes no sense because as used here, these are all terms related to BBB disruption. Different markers and probes can indeed be used in special ways, but this sentence does not help with understanding how.

Response 18: The sentence was now rephrased accordingly:

Ln 796-798 “Both small (for examining small changes) and large molecules (to assess BBB integrity loss) can be used as markers for BBB dysfunction. Thereby, a combination of different markers is most reliable to evaluate the BBB integrity.”

Point 19: Table 1. NMDARE, NMOSD, MOGAD all need to be defined.

Response 19: Although defined in the running text, indeed, we did not define those words within the table. We therefore added definitions for all abbreviations from table 1 additionally below the table for a better overview.

Point 20: Ln 759: Immune cells do not cross the BBB in MS because of “breakdown” or leakage (if that were the mechanism, red blood cells would enter the CNS as well), but because of enhanced diapedesis. So “dysfunction” would be a better word here.

Response 20: We replaced the word “breakdown” with “dysfunction”. (Ln 818)

Point 21: Ln844: SARS-CoV-2 hard to keep track on. I am sympathetic to that, but surely you need to review the article by Rhea given its direct relation to the topic of BBB and its prominent journal.

Response 21: We regret that we have overseen this important input, and added now the following passage to the CNS infections section:

Ln 926-933 “Rhea et al. tested radiolabeled monomeric SARS-CoV-2 spike protein subunit S1 in vivo and in vitro on its ability to cross the BBB. They suggested an adsorptive transcytotic manner with participation of ACE2 in murine brain and lung. Furthermore, they observed a much more efficient uptake of S1 in all brain regions after uptake across the BBB than the nasal route. This uptake of S1 was minimally affected by LPS-induced inflammation, but the inflammation altered S1 clearance from blood and its uptake by peripheral tissues. However, their transwell experiments with human iBMECs showed limited permeability for S1, probably due to technical reasons [272].”

The respective reference is:

  1. Rhea, E.M.; Logsdon, A.F.; Hansen, K.M.; Williams, L.M.; Reed, M.J.; Baumann, K.K.; Holden, S.J.; Raber, J.; Banks, W.A.; Erickson, M.A. The S1 protein of SARS-CoV-2 crosses the blood-brain barrier in mice. Nature neuroscience 2021, 24, 368–378. https://doi.org/10.1038/s41593-020-00771-8.

Point 22: Ln851: “Kase et al. observed that the major SARS-CoV-2 strain spike proteins were able to infect microglia,…” I believe they misquote Kase et al. As written, Kase would have only studied the spike proteins, but they used viral particles to study productive infection.

Response 22: The reviewer is right here, misleadingly Kase et al. called their psydotyped lentiviruses “SARS-CoV-2_S”. “Major SARS-CoV-2 strain spike proteins” was now replaced with “psydotyped lentivirus particles of major SARS-CoV-2 strains”. (Ln 921)

Point 23: Ln 1044-1048: Two sentences for Neurodegeneration just highlights that you have decided not to review the topic in this way. Also, you elsewhere review neurodegenerative diseases such as AD and Parkinson’s. I would suggest you delete this two sentence section.

Response 23: We have deleted both sentences as suggested by the reviewer.

Point 24: Ln 136-7: Both brain resident cells, such as glial cells, and peripheral immune cells that con tribute to neuroinflammation.

Response 24: “that” was removed.

Point 25: Ln 160: “This A1 type astrocyte type…”

Response 25: The second “type” was removed.

Point 26: Ln347: “messes” is vernacular.

Response 26: “messes” was replaced with “interferes”.

Point 27: Ln 381: “…which makes it is crucial…”

Response 27: “is” was removed.

Point 28: Ln 386: “entity”, do you mean “characteristics”?

Response 28: We rephrased the word „entity” with “cellular model”.

Response 29: Ln 702-4: “However, those methods do not reflect the dead cell compartment reductions in a 3D ECM occur only slowly [193]. “

Response 29: An “, and” was forgotten. This sentence was rephrased to

“However, those methods do not reflect the dead cell compartment, and reductions in a 3D ECM occur only slowly.” (Ln 760)

Reviewer 2 Report

Overall I felt this review did an excellent job of introducing readers to the blood-brain barrier and its components while framing it in the context of modeling neuroinflammation. Given that the authors cited over 300 different works they did a formidable job of trying to pull together the literature I do feel that a few groups were missing especially regarding the context of inflammation as I know both a Harvard and a Vanderbilt group have shown blood-brain barrier responses to cytokines and LPS and I saw neither of these groups represented during that section of the discussion. However again there are so many groups at this point it is understandable and no one can accuse the authors of not attempting to do a comprehensive job. I am recommending this review for publication as I do feel it will be an excellent resource for researchers working in this area. 

Author Response

Overall I felt this review did an excellent job of introducing readers to the blood-brain barrier and its components while framing it in the context of modeling neuroinflammation. Given that the authors cited over 300 different works they did a formidable job of trying to pull together the literature I do feel that a few groups were missing especially regarding the context of inflammation as I know both a Harvard and a Vanderbilt group have shown blood-brain barrier responses to cytokines and LPS and I saw neither of these groups represented during that section of the discussion. However again there are so many groups at this point it is understandable and no one can accuse the authors of not attempting to do a comprehensive job.

I am recommending this review for publication as I do feel it will be an excellent resource for researchers working in this area.

Response: We would like to thank the reviewer for this very positive and thoughtful review. We had another look into the literature and have now added the missing references as suggested by you:

Ln 550-554 “The neuroinflammatory response profile to TNF-α stimulation in 2D vs. 3D models was demonstrated by Herland et al. in their mono- and coculture models utilizing transwell and microfluidic BBB-on-a-chip setups. Co-culturing primary BMECs with astrocytes and pericytes, as well as flow conditions, resulted in significantly higher cytokine secretion (IL-6, granulocyte colony-stimulating factor) in the 3D model [70].”

Reference:

  1. Herland, A.; van der Meer, A.D.; FitzGerald, E.A.; Park, T.E.; Sleeboom, J.J.F.; Ingber, D.E. Distinct Contributions of Astrocytes and Pericytes to Neuroinflammation Identified in a 3D Human Blood-Brain Barrier on a Chip. PloS one 2016, 11, e0150360. https://doi.org/10.1371/journal.pone.0150360.

Ln 572-580 “Faley et al. created a 3D microfluidic model with a tubular structure filled with monolayers of endothelial cells. They compared different endothelial cell lines, ranging from human umbilical vein endothelial cells to human dermal microvascular endothelial cells and two iPSC-derived BMEC lines. The iBMECs showed 10-100 times lower permeability to different-sized markers compared to the other cell lines. Moreover, their model maintained barrier function and efflux transporter activity for up to 21 days under perfusion conditions. Worth mentioning is that they saw the best results with a subphysiological wall shear of 0.3 dyn/cm2. Elevated flow rates led to an increase in permeability and angiogenic sprouting [170].”

Reference:

  1. Faley, S.L.; Neal, E.H.; Wang, J.X.; Bosworth, A.M.; Weber, C.M.; Balotin, K.M.; Lippmann, E.S.; Bellan, L.M. iPSC-Derived Brain Endothelium Exhibits Stable, Long-Term Barrier Function in Perfused Hydrogel Scaffolds. Stem cell reports 2019, 12, 474–487. https://doi.org/10.1016/j.stemcr.2019.01.009

Ln 877-887 “Brown et al. investigated the impact inflammatory stimulation on the BBB induced by LPS (mimicking systemic bacterial infection) or pro-inflammatory cytokines (which could result of a local or systemic inflammation) in their 3D microfluidic NVU dual chamber model. By applying either 100 µg/mL of LPS or a cytokine mix with 100 ng/mL (IL-1, TNF-α and MCP-1,2) to the vascular compartment, inflammation was induced. They then looked into the metabolic profile on both sides over the time of 24 h of exposure. Interestingly, they found pro-inflammatory cytokine activation in both vascular and brain compartments, whereas at later time points this activation was only seen in a subset in both compartments. This suggests the activation of pro-repair cytokines at later time points within the brain section, as the vascular compartment remains more pro-inflammatory [76].

Reference:

  1. Brown, J.A.; Codreanu, S.G.; Shi, M.; Sherrod, S.D.; Markov, D.A.; Neely, M.D.; Britt, C.M.; Hoilett, O.S.; Reiserer, R.S.; Samson, P.C.; et al. Metabolic consequences of inflammatory disruption of the blood-brain barrier in an organ-on-chip model of the human neurovascular unit. Journal of neuroinflammation 2016, 13, 306. https://doi.org/10.1186/s12974-016-0760-y.

Ln 996-998 “A 3D microfluidic model by Cho et al. with rat brain endothelial cells was developed to screen BBB-targeting drugs. They induced ischemic conditions by OGD and showed the protective functions of antioxidant edaravone and Rho kinase-inhibitor Y-27632 [71].”

Reference:

  1. Cho, H.; Seo, J.H.; Wong, K.H.K.; Terasaki, Y.; Park, J.; Bong, K.; Arai, K.; Lo, E.H.; Irimia, D. Three-Dimensional Blood-Brain Barrier Model for in vitro Studies of Neurovascular Pathology. Scientific reports 2015, 5, 15222. https://doi.org/10.1038/srep15222.

Round 2

Reviewer 1 Report

The authors have addressed my concerns

The authors have addressed all of my issues.